# STRUCTURED RATIONALE RESPONSES: ENHANCING LANGUAGE MODEL RESPONSES TO NUANCED INQUIRIES IN ELECTRONIC MEDICAL RECORDS

## ABSTRACT

Large language models (LLMs) provide impressive out-of-the-box performance to queries for which data are in the public domain using prompt engineering. However, they are less effective when analyzing important datasets that are not publicly available, such as electronic health records (EHRs), where current prompting strategies are either suboptimal or require domain-specific expertise. To overcome these limitations, we proposed a *non-domain-specific* prompting strategy—termed **Structured Rationale Responses** (SRR)—designed to enhance the accuracy and reliability of LLM responses to nuanced inquiries in EHRs compared with expert interpretations. Specifically, SRR guides LLMs to generate responses 1) in a structured format (*e.g.*, JSON), and 2) with rationales, which are sentences excerpted from the EHR notes that the LLM used to support its answers. In 499 full-text EHR notes (474.6±164.3 words) in 125 patients with life-threatening heart rhythm disorders, we asked LLM whether a patient had an acute event of ventricular arrhythmias which required it to remove parse contradictory information on prior events. In an independent hold-out test set of 398 notes (471.8±160.1 words), our SRR achieved a balanced accuracy of 86.6%±4.0% without any in-context examples, demonstrating an average performance lift of 30.5% over the standard prompts, 12.2% over Zero-shot-CoT prompts, and 10.4% over 5-shot prompts. Notably, for true positives where LLM correctly identified acute events, 94.4%±5.2% had at least one LLM-generated rationale considered clinically relevant by experts. The efficacy and robustness of our SRR were also thoroughly evaluated across different structured formats, LLMs, and in distinct tasks using a public EHR dataset. Our code can be found at `https://github.com/***`.

## 1 INTRODUCTION

The use of large language models (LLM) to interpret natural language datasets has sparked worldwide interest (Yang et al., 2023). These foundation models exhibit remarkable capabilities in processing unstructured data and being generalized to tasks for which they were not originally trained using carefully engineered prompts (Liu et al., 2023b). These achievements are particularly promising in the medical field, where the analysis of complicated, specialized texts has historically relied on time-consuming manual analysis by experts. However, recent attempts to apply LLMs in medicines have unveiled their limitations, particularly in questions requiring higher-order reasoning (Bhayana et al., 2023; Azizi et al., 2023). While several prompting strategies have been developed to increase the reasoning ability of LLMs (Wei et al., 2022; Kojima et al., 2022; Wang et al., 2023; Zhang et al., 2023), most prompt engineering efforts to date have been developed for arithmetic, logical, or commonsense reasoning, making them suboptimal for analyzing medical data such as electronic health records (EHRs). Notably, Singhal et al. (2023) demonstrated significant improvements in medical reasoning, but required domain-specific expertise to align LLMs to the medical domain. Therefore, we seek to answer the following question: *Can we enhance the accuracy and reliability of LLM responses to nuanced inquiries in EHRs with a non-domain-specific prompting strategy?*

To answer this question, we propose a novel prompting strategy, termed Structured Rationale Responses (SRR), to detect recurrent events in EHRs for patients with a known history of ventricular tachycardia. Ventricular tachycardia (VT) or fibrillation (VF) can cause sudden cardiac arrest, a

leading cause of mortality, for which the predictive tools are crucial for specialist referral and resource planning (Chugh et al., 2022). EHRs, as a repository of extensive data, hold significant promise for these purposes; however, few existing tools can accurately interpret EHRs for this nuanced task (Moon et al., 2020; Siontis et al., 2021) because it requires separation of new from prior events and recognition of a diversity of clinical synonyms. Our SRR prompting strategy enables accurate and reliable responses from LLMs, without relying on any domain-specific expertise and, therefore, has the potential to be used by non-experts such as administrators and digital agents for resource planning or patients to better understand their EHRs. To achieve this, our SRR strategy first instructs LLMs to respond in a structured format. We employ JSON in this study. Furthermore, LLMs are guided to not only generate answers in response to a inquiry, but also provide rationales that LLMs use to support their answer.

To evaluate our method, we curated and manually annotated a large cohort of 499 full-text notes containing unstructured, redundant, and sometimes contradictory texts. These notes are typically lengthy, with 474.6±164.3 words, which can potentially reduce LLM performance (Liu et al., 2023a). Furthermore, VT events are relatively uncommon, and the presence of unbalanced classes ($< 7\%$) requires LLMs to pinpoint recurrent events without elevating the rate of false positives.

Our study design is illustrated in Figure 1A. In summary, we make the following four contributions:

- We develop a non-domain-specific prompting strategy to enhance the accuracy and reliability of LLM responses to nuanced inquiries in EHRs.
- We have curated and annotated a large cohort of 499 clinical notes to identify whether a patient with pre-existing history of ventricular arrhythmias experienced a new episode after therapy.
- Our Structured Rationale Responses (SRR) strategy outperformed 4 baseline prompting strategies, including few-shot prompts and Zero-shot-CoT (Kojima et al., 2022), in VT recurrence identification, while the Structured Output Format intervention made a more substantial contribution to this improvement.
- Our Structured Rationale Responses (SRR) strategy generated rationales that closely align with expert interpretations, affirming their reliability and clinical relevance.

## 2 METHODS

Structured Rationale Responses (SRR) is a prompting approach that enhances the accuracy and reliability of LLM responses to nuanced inquiries in EHRs without using domain-specific crafted prompts or in-context examples. It potentially has broader applications and may be useful for patients and non-experts, such as administrators and digital agents, for clinical support or resource utilization.

### 2.1 PRELIMINARIES

We studied whether LLMs could answer a nuanced clinical question. Specifically, we asked whether a patient with a known history of ventricular arrhythmias has had a new episode after therapy. Recognizing new episodes is crucial since they frequently lead to changes in clinical management, including an office evaluation, medication changes, an invasive procedure (ablation) or referral to another center. For this study, we exclusively queried a pre-trained LLM without any further training or fine-tuning.

Suppose we have $\mathcal{N}_i^t \subset \mathcal{T}$, where $\mathcal{T}$ is the *Test cohort* and $\mathcal{N}_i^t$ represents the $i$th test note. Each test note is defined as a collection of sentences formatted as textual strings. The superscript $t$ indicates that the data belongs to the *Test cohort* $\mathcal{T}$. Additionally, $\mathcal{L}_i^t \in \{\text{Yes, No}\}$ is the ground truth of $\mathcal{N}_i^t$ annotated by medical experts. Our goal was to devise a prompt template $\mathcal{P}(\cdot)$ to improve the accuracy of LLM compared to expert judgments, measured by

$$\mathrm{M}\left(\{\mathrm{LLM}\left(\mathcal{P}\left(\mathcal{N}_i^t, \mathcal{Q}\right)\right), \mathcal{L}_i^t\}_{i=1}^{|\mathcal{T}|}\right),$$

where $\mathcal{Q}$ is the clinical question LLM needs to answer, $\mathrm{LLM}(\cdot)$ processes prompts and generates the response, while $\mathrm{M}(\cdot)$ serves as a classification metric measuring the accuracy between LLM-generated responses and expert annotations.

## 2.2 CLINICAL QUESTION

Because slight modifications of the phrasing of a question may influence interpretation by LLMs (Arora et al., 2022), we presented the clinical question in one of broad ($\mathcal{Q}^1$), intermediate ($\mathcal{Q}^2$) and detailed ($\mathcal{Q}^3$) formulations.

$\mathcal{Q}^1$ Has the patient had Ventricular Tachycardia (VT) recurrence after ablation?

$\mathcal{Q}^2$ Has the patient had sustained ventricular tachycardia (VT) or ventricular fibrillation after ablation?

$\mathcal{Q}^3$ Has the patient had sustained ventricular tachycardia (VT) or ventricular fibrillation after ablation? Consider mentioning any episodes of ventricular tachycardia, wide complex tachycardia, sustained or nonsustained VT, and any device interrogation results, such as delivered therapy (shocks, ATP) or VT and VF detection.

$\mathcal{Q}^2$ represents the question posed to clinical experts during the annotation of the notes. $\mathcal{Q}^3$ is automatically refined based on question $\mathcal{Q}^2$ using ChatGPT with GPT-4. This design is inspired by Zhou et al. (2022), which tests whether LLMs have the capability to craft prompts that may equal or exceed the efficacy of those developed by experts.

## 2.3 STRUCTURED RATIONALE RESPONSES

Our Structured Rationale Responses (SRR) strategy includes two interventions: Structured Output Format and Rationale.

**Structured Output Format:** Our first prompt engineering intervention was to ask for a structured output format. Specifically, our SRR regularized the LLM output into a more specific format, *i.e.*, JSON, as opposed to an unstructured text string. While previous study Agrawal et al. (2022) structured LLM responses as a bulleted list, our intervention yields a more rigorous structure, which not only streamlines the post-processing effort required to convert an unstructured response to a label, but also significantly enhances the response accuracy (refer to Section 4.5).

**Rationale:** We observed that answering the aforementioned clinical questions typically requires pinpointing contextual cues within the note. Such cues facilitate recognizing events conveyed through a diversity of clinical synonyms or distinguishing new incidents from historical ones. Hence, our rationale intervention asks the LLM not only to generate an answer to the clinical question but also to provide evidence or rationale, specifically, sentences excerpted from the note that the LLM used to support its answer. Based on our definition of notes in Section 2.1, the rationale $\mathcal{R}_i^t$ will be a subset of note $\mathcal{N}_i^t$, *i.e.*, $\mathcal{R}_i^t \subset \mathcal{N}_i^t$.

Note that Rationale could be delivered in two ways: 1) the LLM first provides its answer followed by the Rationale (Answer $\rightarrow$ Rationale) or, 2) the LLM first provides the Rationale followed by the answer (Rationale $\rightarrow$ Answer). We adopt the latter sequence in our SRR strategy. The comparison of performance with these two sequences are analyzed in Section 4.6.

In summary, giving a free-text note $\mathcal{N}_i^t$ and a clinical question $\mathcal{Q}$, our SRR prompting guides LLMs to first output rationales followed by the answer in a JSON format with two keys: "rationale" and "label." Figure 1B demonstrates our SRR strategy as well as the corresponding LLM response. The detailed prompt definitions are provided in Section A.2 in the appendix.

**Intuition:** Current prompting strategies, such as Chain-of-Thought (Wei et al., 2022) and Zero-shot-CoT (Kojima et al., 2022), aim to enhance LLMs' capabilities through step-by-step logical deductions for tasks involving arithmetic, logical, or commonsense reasoning. However, LLMs often lack adequate domain-specific clinical knowledge for advanced reasoning and clinician-oriented questions (Bhayana et al., 2023; Azizi et al., 2023), primarily due to the scarcity of extensive, diverse EHR datasets in the public domain, which is constrained by data protection regulations like HIPAA (Act, 2023). This limitation can result in unreliable outcomes when LLMs are compelled to reason independently. For instance, as depicted in Figure 1B, Zero-shot-CoT accurately identified a VT-related cue but incorrectly concluded that "wide complex tachycardia > 30 seconds" does not signify a sustained VT event, leading to an erroneous answer (see Section A.4 in the appendix for in-depth analysis). The issue of forced reasoning is further highlighted by Zero-shot-CoT's diminished performance with less sophisticated LLMs (see Section B.2 for more details). In contrast, our

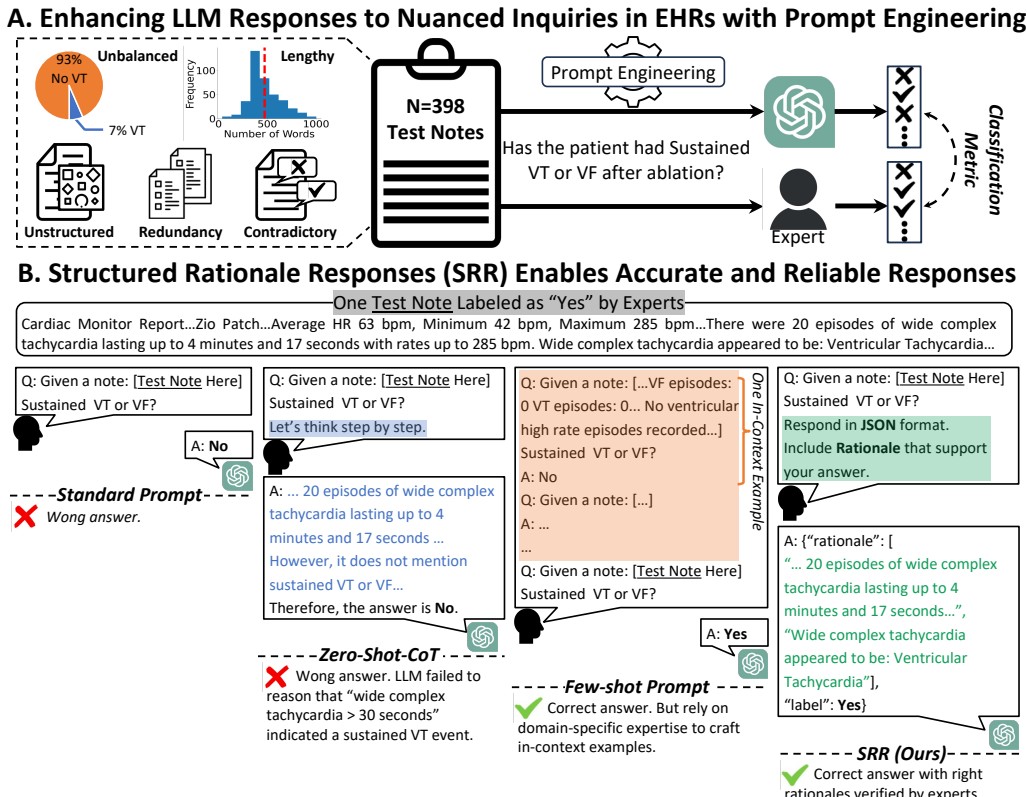

Figure 1: Study Design and Method Illustration. A. This study aimed to devise optimal prompts for LLMs to detect recurrent ventricular tachycardia (VT) post-ablation from medical device notes (N=398) in a hold-out test cohort. Such notes are typically characterized by lengthy, redundant, and contradictory text with an unstructured format and unbalanced labels. B. Our Structured Rationale Responses (SRR) strategy is a prompting approach that enhances the accuracy and reliability of LLM responses by asking LLM to provide supportive evidence for its answer to the clinical question, using sentences from the query note, and output in JSON format. We illustrated Standard Prompts, Zero-Shot-CoT (Kojima et al., 2022), Few-Shot Prompts, and our SRR strategy with example inputs and outputs of an LLM. The full-length test note and example note used in the in-context example are provided in Table 3 in the appendix, numbers 1 and 2, respectively.

Rationale-based intervention, which emphasizes contextual cue identification over explicit deduction, successfully provided the correct answer (Figure 1B). Our approach might lean towards a "Yes" classification in cases of uncertainty, potentially increasing false positives (refer to Section A.4). Despite this, our method's superior ability to identify the minority but critical classes demonstrates its reliability. This attribute is particularly valuable in health applications with highly skewed category distributions, such as coronary heart disease (Janosi & Detrano, 1988), cardiovascular failure, myocardial infarction, or stroke (Bairey Merz et al., 1999). Additionally, requesting structured formatting can diminish ambiguity, allowing LLMs to discern the rationale behind inquiries more rapidly, leading to more effective and efficient computational resource allocation, which is crucial for nuanced tasks.

## 3  STUDY POPULATION AND DATASET CURATION

From the Anonymous Institution VT Registry, we identified N = 125 patients diagnosed with VT who underwent ablation procedures at Anonymous Institution Health Care between 2013 and 2022. For each patient, we identified 3.0 (IQR: 2.0 to 6.0) full-text notes following the date of ablation, with encounter type "Cardiology" or "Cardiovascular". This resulted in a total of 499 notes. We

then stratified the dataset and divided it into N = 25 patients to provide in-context examples for few-shot learning baselines (*Development cohort* $\mathcal{D}$) and N = 100 patients as a hold-out test cohort (*Test cohort* $\mathcal{T}$), ensuring a balance in age, gender, LVEF, device type, and the number of notes. Table 1 in the appendix summarizes the baseline characteristics of all patients.

To protect patient privacy, all notes were first deidentified by removing any protected health information (PHI). We utilized Philter ("Protected Health Information filter") (Norgeot et al., 2020), a recognized open-source deidentification software whose recall on the 2014 i2b2 deidentification challenge (Uzuner & Stubbs, 2015) is the highest reported in the literature (Norgeot et al., 2020). After the automated preprocessing with Philter, the dataset was manually reviewed by the authors for additional assurance.

To annotate the data, clinical experts carefully examined each note to answer one specific question: "Has the patient had sustained ventricular tachycardia (VT) or ventricular fibrillation after ablation?" Then they performed the following tasks: 1) assigned "Yes" or "No" label to each note, and 2) identified sentences within the original note that support their assignment. For terminological clarity, we term these expert-identified sentences as "context phrases", distinguishing them from "rationales" generated by the LLM to justify its responses. We denote the "context phrases" from a note $\mathcal{N}_i^t$ as $\mathcal{C}_i^t$, *i.e.*, $\mathcal{C}_i^t \subset \mathcal{N}_i^t$.

To annotate the dataset, clinical experts meticulously analyzed each medical record to address a specific query: "Has the patient experienced sustained ventricular tachycardia (VT) or ventricular fibrillation post-ablation?" Subsequent tasks involved: 1) assigning a "Yes" or "No" label to each record, and 2) pinpointing sentences in the original record that substantiated their labeling. For clarity in terminology, sentences identified by experts supporting their labels are referred to as "context phrases," distinguished from "rationales" provided by the Language Learning Model (LLM) to rationalize its answers. The "context phrases" from a note $\mathcal{N}_i^t$ are denoted as $\mathcal{C}_i^t$, implying $\mathcal{C}_i^t \subset \mathcal{N}_i^t$, with the superscript $t$ indicating data from the test cohort (*Test cohort* $\mathcal{T}$).

## A. Clinical Note Example of No VT Recurrence and Expert Annotation

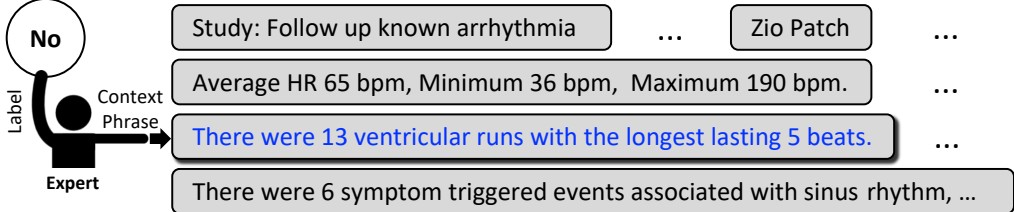

## B. Clinical Note Example of VT Recurrence and Expert Annotation

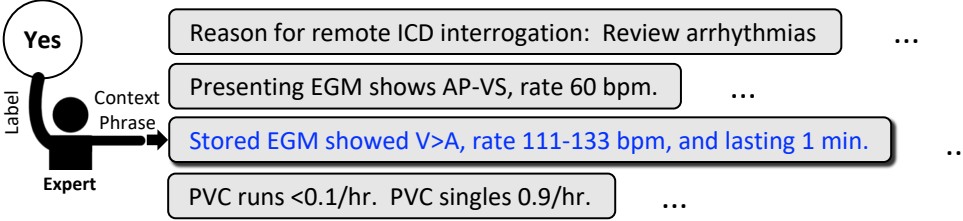

Figure 2: Illustration of de-identified notes. A. A note with VT recurrence that was not clinically significant at only 5 beats. B. A note with VT recurrence, defined by a VT episode lasting 1 minutes. All notes were pre-processed and de-identified. Experts were asked to assign "Yes" or "No" label to each note and to identify context phrases, *i.e.*, sentences within the original note that support their assignment (highlighted in blue). The full-length notes in Panel A and B are provided in Table 3 in the appendix, numbers 3 and 4, respectively.

Figure 2A and 2B showcase notes labeled by experts as "No" and "Yes" (see Table 3 in the appexdix for full notes, numbers 3 and 4, respectively). In Figure 2A, a new VT event occurred but was considered not clinically relevant by experts due to duration 5 beats, and was labeled as "No".

Figure 2B indicates a new episode of VT lasting > 30 seconds that was clinically relevant and coded by experts as "Yes". Overall, the *Development Cohort* $\mathcal{D}$ provided 101 notes with 485.6±179.4 words and 7 notes labeled as "Yes", while our *Test cohort* $\mathcal{T}$ provided 398 notes with 471.8±160.1 words and 26 notes labeled as "Yes".

# 4 RESULTS

## 4.1 EXPERIMENTAL DETAILS

We evaluate our Structured Rationale Responses (SRR) strategy using the hold-out *Test cohort* $\mathcal{T}$ with 398 free-text notes from N = 100 patients. Given the imbalanced label distribution (26/398 positive cases), we employ balanced accuracy (Brodersen et al., 2010) as the classification metric. Additionally, we utilize the Net Reclassification Index (NRI) (Pencina et al., 2008) for statistical analysis to compare the performance of two prompting strategies.

We used ChatGPT API (gpt-3.5-turbo; Open AI, San Francisco, CA) for our experiment. To ensure fair comparisons and reproducibility, we set the temperature parameter to zero in all experiments.

## 4.2 BASELINE PROMPTING STRATEGIES

**Standard Prompts** In the standard prompts, we instruct LLMs to provide a simple "Yes" or "No" response to the clinical question for each test note. No additional guidance or context are provided.

**Zero-Shot-CoT** We implemented the zero-shot chain of thought prompting strategy (Kojima et al., 2022). First, we guide LLMs to reason through the clinical question by prompting "Let's think step by step." Subsequently, we extract the final answer by asking, "Therefore, the answer (Yes or No) is."

**Few-Shot Prompts** In few-shot prompts, we provided LLMs with in-context examples to help learning desired patterns and correct responses. Each in-context examples consists a full-length clinical note along with the corresponding ground truth label assigned by clinical experts. Examples are randomly selected from the *Development Cohort* $\mathcal{D}$ and the experiment is repeated 10 times to assess the consistency of LLM performance across different example selections. We evaluated 1-5 in-context examples. Using more examples will exceed the input limit of 4,097 tokens for the gpt-3.5-turbo model.

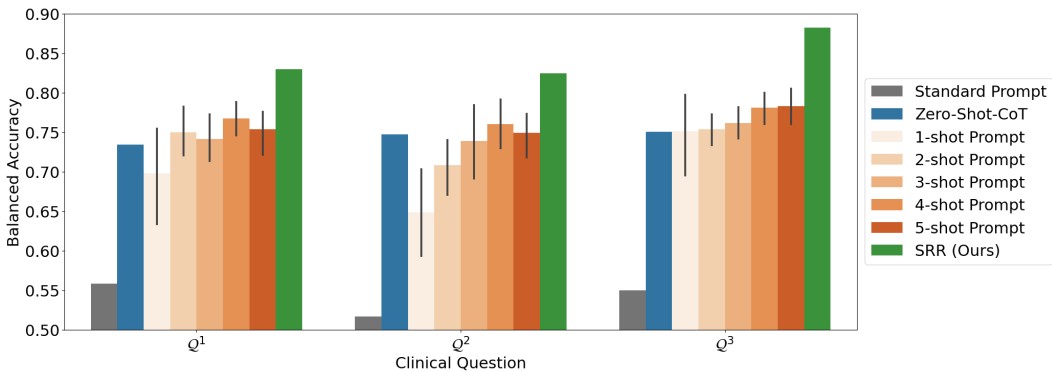

Figure 3: Our SRR Strategy outperforms baseline prompting strategies over 3 clinical questions, demonstrating an average balanced accuracy improvement of 30.5% over the standard prompt (NRI=1.062, $p < 0.001$), 12.2% over Zero-shot-CoT prompt (NRI=0.336, $p = 0.188$), and 10.4% over 5-shot prompt (NRI=0.195, $p = 0.455$).

### 4.3 OUR SRR STRATEGY SHOWED HIGHER BALANCED ACCURACY THAN THE BASELINE PROMPTING STRATEGIES

Figure 3 illustrates the balanced accuracy achieved by different prompting strategies. First, we observed that standard prompts only had a balanced accuracy of 54.1%±2.2% across 3 different clinical questions, failing to identify the positive cases. The Zero-Shot-CoT prompting strategy guided LLMs to reasoning before answering, resulting in an improved balanced accuracy of 74.5%±0.9%. However, as demonstrated in Figure 1B, Zero-Shot-CoT was capable of recognizing context cues such as "wide complex tachycardia" but failed to reason that "wide complex tachycardia > 30 seconds" indicated a sustained VT event.

For few-shot prompts, we observe adding one in-context example improved balanced accuracy to 69.9%±10.2%. The addition of more examples led to a slight improvement in mean accuracy and a decrease in variances, but the improvement is marginal after the first example (NRI=0.269, $p = 0.251$, 1 vs 5 examples), as 5-shot prompt achieves a balanced accuracy of 76.2%±4.4%. It is important to note that these improvements rely on domain-specific expertise for manually crafting the in-context examples.

Finally, our SRR strategy achieved a balanced accuracy of 84.6%±3.2%, demonstrating an average performance lift of 30.5% over the standard prompt (NRI=1.062, $p < 0.001$), 12.2% over Zero-shot-CoT prompt (NRI=0.336, $p = 0.188$), and 10.4% over 5-shot prompt (NRI=0.195, $p = 0.455$). This substantial performance improvement demonstrated the effectiveness of our proposed non-domain-specific prompt strategy.

### 4.4 OUR SRR STRATEGY GENERATED RELIABLE RATIONALES COMPARED TO EXPERT INTERPRETATIONS

Given the superior classification performance of our SRR strategy, a critical question one should ask is: "*Can LLM identify the correct context cues or rationales to support its answers?*" To answer this question, we conducted a comparison between the rationales ($\mathcal{R}_i^t$) generated by LLM for test note $\mathcal{N}_i^t$ and the corresponding context phrases ($\mathcal{C}_i^t$) identified by clinical experts. We calculated the ratio of test cases in which at least one LLM-generated rationale matched an expert-identified context phrase. Mathematically, this can be expressed as: $|\{\mathcal{N}_i^t \subset \mathcal{T}|\mathcal{R}_i^t \cap \mathcal{C}_i^t \neq \emptyset\}|/|\mathcal{T}|$. However, achieving a perfect match between LLM-generated rationales and expert-identified context phrases can be challenging in practice due to potential differences in sentence segmentation. To address this, we defined a rationale as matching a context phrase if at least *85% of consecutive characters overlapped* between the rationale and the context phrase.

We found out that the ratio is remarkable 94.4%±5.2% for true positives and 77.9%±6.3% for all test cases three clinical questions. Figure 4 showed two notes with highlighted rationales and context phrases. It is noteworthy that the former is more important since the identification of patients with VT/VF recurrent events is crucial for arrhythmia management, specialist referrals, and resource planning. The inclusion of rationales using our SRR strategy significantly enhanced the reliability of LLM-generated responses and offers an simple way for clinical experts to verify these responses.

### 4.5 STRUCTURED OUTPUT FORMAT EXHIBITED MORE PRONOUNCED IMPACT IN CLASSIFICATION ACCURACY THAN RATIONALE

In this section, we conduct an ablation study on our SRR strategy, focusing on the impact of two key interventions: Structured Output Format and Rationale. Specifically, with Structured Output Format intervention, we prompted LLM to respond in a JSON format with one key: "label". And with the Rationale intervention, we asked LLMs to provide a bulleted list of rationales, followed by a "Yes" or "No" answer. Both strategies are evaluated within the same experimental framework as previously described. The experimental results are presented in Figure 5. The detailed prompt definitions utilized in this experiment can be found in Section A.2 in the appendix.

By adopting the Structured Output Format alone, LLMs achieved an average accuracy of 80.1%±4.5% across three clinical questions, with a significant improvement of 26.0% compared to the standard prompt (NRI = 0.898, $p < 0.001$). However, the Rationale strategy yielded a balanced accuracy of only 60.4%±2.8%.

**Note 1**

Cardiac Monitor Report Reason for Study: Ventricular tachycardia Type of cardiac monitor:  Zio Patch Cardiac monitor was worn for 14 days and 0 hours. Interpretation Summary 1. Recording quality: Good 2. Sinus Rhythm was observed.  Average HR 63 bpm, Minimum 42 bpm,  Maximum 285 bpm…5. Atrial fibrillation and/or flutter was not present.   6. PVC burden was: Isolated Frequent 5.3%, Couplet and Triplet rare <1.0% There were 20 episodes of wide complex tachycardia lasting up to 4 minutes and 17 seconds with rates up to 285 bpm. Wide complex tachycardia appeared to be: Ventricular Tachycardia 7. There were no patient triggered events. There were no patient symptom episodes. IMPRESSION:  Results demonstrate predominantly sinus rhythm….

**Note 2**

Reason for remote ICD interrogation: Review arrhythmias, Evaluate lead function and Check battery status Observations/Comments…The histogram demonstrated a normal heart rate profile…23 mode switch episodes recorded…1 treated VT event recorded *** at 11:37 PM).  Stored EGM showed sudden onset V>A, rate 143-188 bpm, ATP x 1 given, rate continued at 194-240 bpm for 9 beats, then terminated (type 2 break). Last treated VT event was on *** (also terminated by ATP x 1). 54 NSVT events recorded. Stored EGMs showed V>A, rate 128-200 bpm, and lasting 6-12 beats. 30 VSEs recorded…There was a 2.3% AT/AF burden. There was one episode of VT requiring ATP which was successful.…Normal impedance, sensing and pacing thresholds. Normal device function.

Figure 4: Our SRR Strategy generates rationales that closely align with expert interpretations, affirming their reliability and clinical relevance. We illustrated two cases of VT recurrence classified accurately by SRR in response to clinical question $\mathcal{Q}^3$. The SRR-generated rationales are marked in green, while the expert-annotated context phrases are highlighted in red. The two full-length notes are provided in Table 3 in the appendix, numbers 1 and 5, respectively.

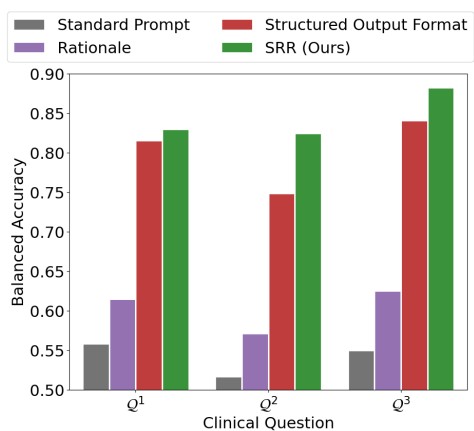

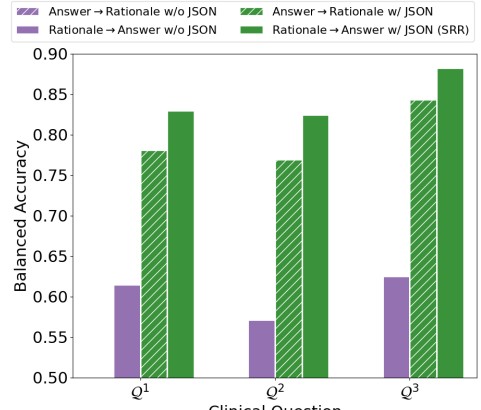

Figure 5: An ablation study on the impact of two key interventions on our SRR strategy. Structured Output Format made a more substantial contribution to enhancing classification accuracy compared to the Rationale.

Figure 6: Rationale before the answer trumps the alternative way, regardless of whether the LLM response is unstructured or structured. LLM cannot identify any positive cases using Answer→Rationale in unstructured responses.

## 4.6 RATIONALE BEFORE THE ANSWER OUTPERFORMED THE ALTERNATIVE SCENARIO

As we mentioned in Section 2.3, Rationale could be delivered via two ways: 1) LLM answers then Rationale (Answer→Rationale) or, 2) Rationale then LLM answer (Rationale→Answer). We tested the two directions without/with using JSON format.  Figure 6 demonstrates the results.

Notably, rationale before the answer outperformed the alternative scenario both without JSON (50.0%±0.0% (failed to detect any positives) vs 60.4% ± 2.8%; NRI=0.211, $p = 0.061$) and with JSON (79.8%±4.0% vs 84.6%±3.2%; NRI=0.251, $p = 0.335$). This outcome aligns with human cognition in which engaging in rational thinking prior to responding enables an individual to better process a question, and echoes insights from the chain-of-thought prompting (Wei et al., 2022) and Zero-Shot-CoT (Kojima et al., 2022) to enhance the efficacy of LLMs in complex arithmetic symbolic reasoning.

### 4.7 OUR SRR STRATEGY IS EFFECTIVE AND ROBUST ACROSS DIFFERENT STRUCTURED FORMATS, LLMS, AND TASKS

The efficacy and robustness of our SRR strategy were thoroughly evaluated across different structured formats, such as YAML and INI (Section B.1), with a different LLM (Section B.2), and in s distinct task using a public EHR dataset (Section B.3).

First of all, we showed using structured formats, such as YAML and INI, either standalone or in conjunction with rationale (*i.e.*, SRR), markedly enhanced performance over the baseline. Notably, SRR using YAML format significantly outperforms YAML alone (NRI = 0.736, $p = 0.001$).

Second, when employing a LLM fundamentally different from GPT-3.5-turbo, SRR significantly surpassed the standard prompt (NRI = 1.131, $p < 0.001$) and Zero-Shot-CoT (NRI = 0.925, $p < 0.001$). It also demonstrated a similar performance to the 5-shot prompt (NRI = -0.125, $p = 0.640$), but without the need for expert annotations.

Lastly, we applied our SRR strategy on a different task: smoking status classification in the 2006 n2c2 De-identification and Smoking Status Challenge (Uzuner et al., 2008). Despite the substantial difference in the task and data, our SRR maintained superior performance compared to the standard prompt, and achieved better macroaveraged F1 score and Cohen's kappa ($\kappa$) compared to Zero-Shot-CoT.

## 5 RELATED WORK

### 5.1 PROMPT ENGINEERING

Prompt engineering has gained considerable attention (Liu et al., 2023b), which aims to optimize "out-of-box" LLM performance on specific tasks, either through altering the formulation of the questions, providing more guidance, or integrating in-context examples. Due to its efficiency in terms of data and time, prompt engineering has become a widely adopted method for various public-domain natural language processing (NLP) tasks (Sanh et al., 2021; Zhou et al., 2022; Ouyang et al., 2022). One of the major interests in this area is to guide LLMs to explicitly generate step-by-step free-text rationales to explain their answer, improving the arithmetic, logical, or commonsense reasoning ability Wei et al. (2022); Kojima et al. (2022); Wang et al. (2023); Zhang et al. (2023). However, we observe this type of "rationale" is suboptimal when addressing nuanced inquiries in EHRs. Instead, our SRR strategy is designed to identify contextual cues within EHRs, resulting in a substantial performance improvement compared to Kojima et al. (2022).

### 5.2 NATURAL LANGUAGE PROCESSING IN MEDICINE

The complexities of medical notes, coupled with their private, relatively small nature (thousands of words unlike petabytes on the internet), make it more difficult to build NLP tools in the field of medicine. For example, Turchioe et al. (2022) reviewed 37 NLP studies focused on cardiac disease, and only one rule-based tool specifically addressed VT and VF Moon et al. (2020). These rule-based NLP tools (Moon et al., 2020; Siontis et al., 2021) are typically based on identifying named entities and keywords using regular expression ("regex") and constructing task-specific rules or logics, which are challenging to differentiate the context and implication of "recurrent VT" from any mention of the term "VT", such as notes referring to recurrent VT before but not after ablation procedures. Recent work such as Singhal et al. (2023) and McInerney et al. (2023) have employed LLMs, which have better semantic analysis ability in medical tasks, but require domain-specific

expertise to optimize prompts or specify high-level features. In contrast, our SRR strategy is a non-domain-specific LLM prompting method, which is more generalizable to different tasks and domains.

# 6 DISCUSSIONS

We proposed a Structured Rationale Responses (SRR) prompting strategy, which can accurately identify patients with recurrent VT after ablation—a nuanced clinical question—from free-text clinical notes. The SRR approach is capable of generating structured responses that include identified contextual cues or rationales of recurrent events, which are closely aligned with expert interpretations. The robustness and effectiveness of SRR were validated across various structured formats and LLMs, as well as through a task utilizing a public EHR dataset. Being non-domain-specific, our SRR strategy can potentially be adapted for broad applicability in different tasks and domains, laying the groundwork for automated analysis of large medical data repositories, which may help to improve medical decision-making across various medical conditions.

**Clinical Data Annotation** In adherence to ethical guidelines, prior to commencing this study, signed consent was obtained from each participant, and access to their health records was authorized by the Anonymous Institution Institutional Review Board (IRB). To ensure confidentiality, all PHI was removed from the clinical notes, which were then independently annotated by a clinical expert with an adequate medical background. We recognize that relying on a single expert's judgement could introduce subjective biases, potentially leading to annotation inconsistencies. To mitigate this, we are in the process of assembling a group of experts to annotate our dataset, aiming to enhance the reliability and consistency of the annotations and facilitate robust inter- and intra-observer agreement analysis. Furthermore, the involvement of multiple experts allows for a more comprehensive and balanced understanding of the data, which is crucial in clinical research to ensure that findings are not only accurate but also reflective of varied medical perspectives.

**Risk of Processing Clinical Data with LLMs** Prior to sending clinical data to LLMs via Chat-Bots or APIs, careful consideration is imperative. Recent research has demonstrated the potential for reconstructing training data from LLMs, even when concealed behind APIs (Kim et al., 2023; Lukas et al., 2023; Huang et al., 2022), posing a risk of privacy leakage. To counteract these risks, it is essential to prevent the use of input clinical data for training subsequent models, a criterion applied in selecting LLMs for our study (Section B.2). Additionally, implementing rigorous data anonymization protocols is crucial. A more conservative approach involves avoiding human review of the data, even for abuse detection purposes[1]. On the other hand, the generic nature of LLMs often fails to capture the nuances and specificities of medical language, raising significant concerns about the reliability and validity of their outputs in clinical settings. Therefore, verifying model outputs with expert interpretations is important. Our SRR offers a distinct advantage in this regard by providing clear rationales to support its answers, which are more readily verifiable by experts compared to extensive notes. Additionally, generating rationales based on the original notes can mitigate the risk of "hallucination" by anchoring the LLM's responses in source data.

**Limitations and Future Work** This study curated and labeled a dataset with a narrowly focused range of encounter types "Cardiology" or "Cardiovascular" from N=125 patients at a single center. Future work should expand the study cohort by incorporating data from multiple institutions and including a broader spectrum of note types. The upcoming analysis and labeling should involve an adjudication committee, including external experts, and consider different medical conditions and comorbidities. Moreover, the evaluations only utilized general-domain LLMs, while future investigations should pivot towards domain-specific, yet smaller, language models, particularly those pre-trained on EHR data (Alsentzer et al., 2019; Kraljevic et al., 2021). This shift is expected to yield insights to facilitate the development of models tailored to the nuanced complexities of medical data, potentially leading to advancements in personalized medicine and predictive analytics in healthcare. Finally, the conducted experiments predominantly focused on English text, constrained largely by the availability of relevant datasets. While the specifics of our results may differ across different languages, the fundamental intuition behind our SRR strategy will likely hold. Additionally, the adaptation of our evaluation framework to corpora in diverse languages is not only feasible but can significantly expand the scope of potential research. This expansion is enabled by the inherent

---

[1]Responsible use of MIMIC data with online services like GPT

multilingual capabilities of modern LLMs, representing a significant advantage over traditional NLP methodologies.

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

## A  APPENDIX

### A.1  BASELINE CHARACTERISTICS OF STUDY COHORT

Table 1 shows the baseline characteristics.

Table 1: Baseline Characteristics

| Characteristics | | *Study Cohort* N=125 | *Development Cohort* $\mathcal{D}$ N=25 | *Test Cohort* $\mathcal{T}$ N=100 | p-values |
|---|---|---|---|---|---|
| Patient Characteristics | | | | | |
| Age, yrs | | 60.6 ± 13.4 | 61.1 ± 14.0 | 60.5 ± 13.2 | 0.84 |
| Sex | Male | 85 (68.0%) | 18 (72.0%) | 67 (67.0%) | 0.81 |
| | Female | 40 (32.0%) | 7 (28.0%) | 33 (33.0%) | |
| Device Type | ICD Device | 59 (47.2%) | 12 (48.0%) | 47 (47.0%) | 1.0 |
| | Cardiac Patch | 66 (52.8%) | 13 (52.0%) | 53 (53.0%) | |
| Left Ventricular Ejection Fraction (%) | | 48.9±13.9 | 50.0±11.7 | 48.6±14.4 | 0.65 |
| Num of Notes | | 3.0 (2.0–6.0) | 3.0 (2.0–5.0) | 3.0 (2.0–6.0) | 0.92 |
| Note Characteristics | | | | | |
| VT After Ablation | Yes | 33 (6.6%) | 7 (6.9%) | 26 (6.5%) | 1.0 |
| | No | 466 (93.4%) | 94 (93.1%) | 372 (93.5%) | |
| Number of Tokens/Words per Note | | 474.6±164.3 | 485.6±179.4 | 471.8±160.1 | 0.45 |
| Number of Characters per Note | | 1371.0±485.0 | 1390.2±524.1 | 1366.2±474.4 | 0.66 |

## A.2 DETAILED PROMPT DEFINITIONS FOR VT RECURRENCE IDENTIFICATION TASK

Table 2 shows the detailed prompt definitions used in the main text.

| |
|---|
| **Standard Prompt** |
| ### Question: |
| Based on the given medical note as input, <Clinical Question $\mathcal{Q}^1$, $\mathcal{Q}^2$, or $\mathcal{Q}^3$ Here>. |
| ### Input: |
| <Test Note Here> |
| ### Response: |
| **Zero-shot-CoT Prompt** |
| ### Question: |
| <Same as Standard Prompt> |
| ### Input: |
| <Test Note Here> |
| ### Response: |
| Let's think step by step. |
| **Our SRR Prompt** |
| ### Question: |
| <Same as Standard Prompt> |
| ### Input: |
| <Test Note Here> |
| ### Instruction: |
| Above is a question, paired with an input that provides further context. Write a response in JSON format that answers the question. The response should be a dictionary with two keys: "rationale" and "label". The value of the "rationale" key should be a bulleted list of sentences supporting the answer of the "label". Each sentence should be a quote from the input medical note. The value of the "label" key should be "Yes" or "No". Your response should only contain the JSON string. |
| ### Response: |
| **Structured Output Format Prompt** |
| ### Question: |
| <Same as Standard Prompt> |
| ### Input: |
| <Test Note Here> |
| ### Instruction: |
| Above is a question, paired with an input that provides further context. Write a response in JSON format that answers the question. The response should be a dictionary with one key: "label". The value of the "label" key should be "Yes" or "No". Your response should only contain the JSON string. |
| ### Response: |
| **Rationale Prompt** |
| ### Question: |
| <Same as Standard Prompt> |
| ### Input: |
| <Test Note Here> |
| ### Instruction: |
| Above is a question, paired with an input that provides further context. Write a response that appropriately answers the question. Start with a bullet-point list of sentences from the medical note that support your answer. Conclude with either "Yes" or "No". |
| ### Response: |

Table 2: Prompt Definitions for the VT Recurrence Identification Task Used in the Main Text.

## A.3 ILLUSTRATIONS OF FULL-LENGTH NOTES

Table 3 illustrates the full-length notes referred to in the main manuscript.

Table 3: The full-length notes referred to in the main manuscript

| Note Number | Reference in the Main Manuscript | Full-length Notes | VT Recurrence |
|---|---|---|---|
| 1 | Figure 1: Test Note & Figure 4: Note 1 | NARRATIVE: ***, *** *** 2:42 PM Cardiology Laboratory *** *** Care Name: *** MD Age: 62 Y MRN: *** DOB: *** Beginning Study Date: *** Cardiac Monitor Report Reason for Study: Ventricular tachycardia Type of cardiac monitor: Zio Patch Cardiac monitor was worn for 14 days and 0 hours. Interpretation Summary 1. Recording quality: Good 2. Sinus Rhythm was observed. Average HR 63 bpm, Minimum 42 bpm, Maximum 285 bpm. 3. There were no pauses. AV Block (2ndÂ° Mobitz II, 3rdÂ°) was not present. 4. Supraventricular Ectopic (SVE) beat burden was: Isolated, Couplet, and Triplet rare <1.0% There was 1 episode of SVT up to 4 beats. The fastest rate was 110 bpm. SVT appeared to be: Atrial Tachycardia 5. Atrial fibrillation and/or flutter was not present. 6. PVC burden was: Isolated Frequent 5.3%, Couplet and Triplet rare <1.0% There were 20 episodes of wide complex tachycardia lasting up to 4 minutes and 17 seconds with rates up to 285 bpm. Wide complex tachycardia appeared to be : Ventricular Tachycardia 7. There were no patient triggered events. There were no patient symptom episodes. IMPRESSION: Results demonstrate predominantly sinus rhythm. ACCESSION NUMBER: *** | Yes |

| 2 | Figure 1: Note in the first in-context example | Reason for remote ICD interrogation: Review arrhythmias, Evaluate lead function and Check battery status Device Manufacturer: Medtronic Model: DDMB1D4 Evera MRI XT DR Retrieved Data: Atrial Paced: 19.95%, Ventricular Paced: 0.11% VF episodes: 0 VT episodes: 0 NSVT/SVT episodes: 0 Number of Mode Switch Episodes: 4 Atrial High Rate Episodes: 1 Battery Status: Battery OK at 2.99 V with estimated longevity of 7.1 years. Charge time: 3.703 seconds Device Measured Data: Atrial lead: Measured Impedance: 418 ohms Measured P Wave: 1 mV Measured Threshold : 0.5 V at 0.4 ms Right Ventricular lead: Measured Impedance: 399 ohms Measured R Wave: 11.75 mV Measured Threshold: 1.375 V at 0.4 ms RV defibrillation impedance: 53 ohms Programmed Parameters: Brady Parameters: Mode: AAIR DDDR . Lower Rate: 50 bpm Upper Track Rate: 130 bpm Upper Sensor Rate: 120 bpm Mode Switch Rate: 171 bpm Atrial lead: Sensitivity: 0.3 mV Output: 1.5 V at 0.4 ms. RV lead: Sensitivity: 0.3 mV Output: 3 V at 0.4 ms. Observations/Comments Scheduled ICD remote transmission received and reviewed. EGM today shows As-Vs at 60's bpm. The lead sensing signals, auto-captured thresholds and impedances are stable. Patient has had 4 mode switch episodes and 1 AT/AF episode recorded, and longest episode lasting 3 minutes ( AT/AF burden $<$ 0.1% of the time). Stored marker channels show A>V with atrial rate at 200 bpm and ventricular rate 63-130 bpm, with frequent PVCs seen. Patient is on Apixaban. No ventricular high rate episodes recorded. 2 SVT-ST episodes recorded, with longest episode lasting 5 minutes. Stored EGMs show 1:1 A=V at 160's bpm. The histogram demonstrated a normal heart rate profile. No short V-V events. PVC runs from 26.2/hr to 9.5/hr. PVC singles from 594.0/hr to 286.2/hr. Normal ICD function. Left VM stating remote transmission has been received and to call the clinic for any questions. Physician Attestation I reviewed the nursing note and agree with the documented findings on interrogation and plan of care. No sustained events recorded. Normal impedance, sensing and pacing thresholds. Normal device function. PVC counts improving. | No |
|---|---|---|---|
| 3 | Figure 2A | Cardiology Laboratory *** Care Name: *** Age: 59 Y MRN: *** DOB: *** Study Date: *** Cardiac Monitor Report Reason for Study: Follow up known arrhythmia Type of cardiac monitor: Zio Patch Interpretation Summary IMPRESSION: Cardiac monitor was worn for 11 days 6 hour(s). Results demonstrate predominant rhythm is sinus with average HR 65 bpm, Minimum 36 bpm, Maximum 190 bpm. There were no supraventricular runs. There were 13 ventricular runs with the longest lasting 5 beats. There were 6 symptom triggered events associated with sinus rhythm, ventricular bigeminy and PVC's. I have personally reviewed the results of this test and agree with the report above. | No |

| 4 | Figure 2B | Reason for remote ICD interrogation: Review arrhythmias, Evaluate lead function and Check battery status Device Manufacturer: Medtronic Model: DDBB1D1 Evera XT DR Retrieved Data: Atrial Paced: 20.81%, Ventricular Paced: 0.08% VF episodes: 0 VT episodes: 0 NSVT episodes: 2 SVT episodes: 0 Mode Switch Episodes: 1 Battery voltage: 2.97 V Charge time: 3.933 seconds Estimated Longevity: 4.2 years Device Measured Data: Atrial lead: Measured Impedance: 513 ohms Measured P Wave: 2.125 mV Measured Threshold: 0.875 V at 0.4 ms (AUTOCAPTURE ON MONITOR) Right Ventricular lead: Measured Impedance: 342 ohms Measured R Wave: 6.875 mV Measured Threshold: 0.75 V at 0.4 ms RV defibrillation impedance: 42 ohms SVC defibrillation impedance: 52 ohms Programmed Parameters: Brady Parameters: Mode: AAI<=>DDD Lower Rate: 60 bpm Upper Track Rate: 110 bpm Upper Sensor Rate: 120 bpm Mode Switch Rate: 171 bpm Atrial lead: Sensitivity: 0.3 mV; Output: 2.5 V at 0.6 ms. RV lead: Sensitivity: 0.3 mV; Output: 2 V at 0.4 ms. Tachy Parameters: VF: 188 bpm FVT via VT: 162 bpm VT: 133 bpm Monitor: 122 bpm Observations/Comments Scheduled ICD remote transmission received and reviewed. Presenting EGM shows AP-VS, rate 60 bpm. Since ***, A-Pace 20.81%, V-Pace 0.08%. The lead sensing signals, capture thresholds and impedances are stable. No short V-V intervals. The histogram demonstrated a normal heart rate profile. Battery OK at 2.97 V with estimated longevity of 4.2 years. 1 mode switch episode, lasting <1 min, but no EGM for review. No atrial high rate events recorded. 1 monitored VT event recorded on ***. Stored EGM showed V>A, rate 111-133 bpm, and lasting 1 min. 1 NSVT recorded, which is the onset for the VT event. PVC runs ¡0.1/hr. PVC singles 0.9/hr. Report to be reviewed by Dr. ***. Next device check on *** in conjunction w/ pt's clinic visit w/ ***, NP. Normal ICD function. Confirmed w/ pt receipt of remote transmission. Pt unable to recall VT event. ATTENDING ***, MD, have interpreted the device check. As a result of this read, the findings are NORMAL DEVICE FUNCTION. | Yes |
|---|---|---|---|

| 5 | Figure 4: Note 2 | Reason for remote ICD interrogation: Review arrhythmias, Evaluate lead function and Check battery status Observations/Comments CRT ICD remote transmission received on *** and reviewed on *** Presenting EGM on *** showed AP-BIV paced, rate 55 bpm. Since ***, A-Pace 76.9%,CRT-Pace 99.1% (of which, VSR pacing is 0.1%). The histogram demonstrated a normal heart rate profile. The lead sensing signals, capture thresholds, and impedances are stable. No short V-V intervals. Battery OK at 2.97 V (charge time 7.3 secs) with estimated longevity of 4.8 years. 23 mode switch episodes recorded. AT/AF burden 2.3% from *** to ***. V rate during AT/AF mostly between 50-110 bpm per histogram (minimally up to 150 bpm). Longest events occurred on *** (14.6 hrs), *** (4.6 hrs), and on *** (19.8 hrs). Medication profile includes rivaroxaban and sotalol. Pt s/p DCCV on *** 1 treated VT event recorded *** at 11:37 PM). Stored EGM showed sudden onset V>A, rate 143-188 bpm, ATP x 1 given, rate continued at 194-240 bpm for 9 beats, then terminated (type 2 break). Last treated VT event was on *** (also terminated by ATP x 1). 54 NSVT events recorded. Stored EGMs showed V>A, rate 128-200 bpm, and lasting 6-12 beats. 30 VSEs recorded. #617 and #619 also recorded in episode list. 3 stored events showed V>A, rate 105-143 bpm, and lasting 5-20 secs (likely NSVT). #615 showed V>A, rate 80's-100's, and lasting 6 secs. #614 showed A>V w/ irregular VS, rate 113-188 bpm, and lasting 5 beats (likely AF w/ RVR). PVC runs 0.6/hr and PVC singles 35.9/hr. Normal CRT ICD function. Report to be reviewed by Dr. ***. Pt has a video visit w/ ***, NP on *** Next remote transmission on *** Sent MHM confirming receipt of remote transmission and to call the Device Clinic at *** for any questions. Physician Attestation I reviewed the nursing note and agree with the documented findings on interrogation and plan of care. There was a 2.3% AT/AF burden. There was one episode of VT requiring ATP which was successful. On anticoagulation. Will continue to monitor for now, on sotalol. Normal impedance, sensing and pacing thresholds. Normal device function. | Yes |
|---|---|---|---|

### A.4 Error Analysis of Responses Using Zero-Shot-CoT and Our SRR

We conducted an in-depth analysis of the responses generated by the LLM using Zero-Shot-CoT and our SRR using the clinical question $\mathcal{Q}^3$, illustrated in Table 4. We studied 6 cases, which included 3 VT-recurrence cases and 3 non-VT-recurrence cases. Markably, our SRR demonstrated a consistent ability to accurately identify context cues aligned with expert-identified context phrases, irrespective of the final answer's correctness. This observation reinforces our findings in Section 4.4. Additionally, SRR tended to yield a "Yes" response in more nuanced cases, such as cases 4 and 5, where "wide complex tachycardia lasting up to *10 beats*" was deemed clinically irrelevant by the expert.

In contrast, Zero-Shot-CoT often produced extensive analyses that were challenging for clinicians to interpret, unlike the more concise rationales generated by SRR. Furthermore, Zero-Shot-CoT exhibited a tendency to respond with "No", even when relevant context cues were present in its step-by-step reasoning (Cases 1-3). This led to a higher incidence of false negatives, potentially resulting in significant delays in clinical intervention or resource allocation due to misdiagnoses. However, Zero-Shot-CoT accurately recognized ATP events mentioned in the notes as historical in Case 6, a task at which our SRR strategy did not succeed. This highlights the importance of common sense reasoning when used properly, and a potential area for improvement in our SRR approach. Future work should explore how to effectively amalgamate common sense reasoning with the structured rationale provided by SRR to enhance its reliability and applicability in complex clinical tasks.

Table 4: Error analysis of responses using Zero-Shot-CoT and our SRR with the clinical question $Q^3$. To increase the readability, we removed the JSON format in SRR responses. The context phrases identified by the clinical expert are highlighted in red. The correct responses are highlighted in green.

| **Case 1 (Yes VT Recurrence)** |
|---|
| Interrogation completed...The patient's underlying rhythm is Sinus 60's...The following therapy was delivered: 0 shocks and 1 ATP. Patient has had 1 episodes of VT and 0 episodes of VF detected....VT monitor from 120 bpm to 115 bpm. UTR from 115 bpm to 110 bpm. Summary Comments:...1 High ventricular events. Review of EGM shows sudden onset, VS>AS, at rate of 150 bpm, with ATP x1 that converts rhythm....Normal ICD function. |

| **SRR Response** | **Zero-Shot-CoT Response** |
|---|---|
| rationale:
- 1 VT episode detected
- 1 ATP delivered
- Review of EGM shows sudden onset, VS>AS, at rate of 150 bpm, with ATP x1 that converts rhythm
- Normal ICD function
label: **Yes** | Based on the given medical note, the patient has had 1 episode of ventricular tachycardia (VT) detected. The patient has not had any episodes of ventricular fibrillation (VF) detected. The device interrogation results show that the patient has received 0 shocks and 1 ATP (anti-tachycardia pacing) therapy. The therapy was delivered for the episode of VT, which had a sudden onset at a rate of 150 bpm and was successfully converted with ATP. The patient was unable to correlate this event with any symptoms. The device check report confirms that the device status is normal and all findings and interpretations have been accurately documented. The programming changes made include adjusting the VT monitor zone from 120 bpm to 115 bpm and the upper tracking rate (UTR) from 115 bpm to 110 bpm. The VT zone with therapy is set at 136 bpm. Overall, there is evidence of an episode of VT, but no sustained ventricular tachycardia or ventricular fibrillation after ablation. Therefore, the answer is No. |

| **Case 2 (Yes VT Recurrence)** |
|---|
| Interrogation completed...Patient has had 0 episodes of VT and 0 episodes of VF detected...Summary Comments:...Available EGM showed A>V with ventricular rate mostly in 70's-90's bpm and minimally in 90's-140's bpm (per histogram) and longest episode lasting 40+ hours (current episode commenced on *** at 1:31 pm and is the longest episode)...2 ventricular high rate events recorded, one as NSVT and one as VT mon. Stored EGM for NSVT episode on *** showed sudden onset with Vs, V¿A, rates 146-182 bpm, lasting 5 beats. Stored EGM for VT mon episode on *** showed unclear onset, A=V 1:1, rates in 120's bpm with eventually A falling in refractory leading to Ap; morphology matches intrinsic about half of the time. Episode lasted 7 minutes. Pt asymptomatic during this time...Monitored VT in 120s... |

| **SRR Response** | **Zero-Shot-CoT Response** |
|---|---|
| rationale:
- Stored EGM for NSVT episode on *** at 2:33 pm showed sudden onset with Vs, V>A, rates 146-182 bpm, lasting 5 beats.
- Stored EGM for VT mon episode on *** at 4:46 am showed unclear onset, A=V 1:1, rates in 120's bpm with eventually A falling in refractory leading to Ap; morphology matches intrinsic about half of the time.
- Episode lasted 7 minutes.
label: **Yes** | Based on the given medical note, the patient has not had sustained ventricular tachycardia (VT) or ventricular fibrillation (VF) after ablation. Here are the reasons why:
1. The patient has had 0 episodes of VT and 0 episodes of VF detected.
2. The stored EGM for the VT episode on *** at 4:46 am showed unclear onset, A=V 1:1, rates in 120's bpm with eventually A falling in refractory leading to Ap; morphology matches intrinsic about half of the time. This episode lasted 7 minutes, and the patient was asymptomatic during this time.
3. The stored EGM for the NSVT episode on *** at 2:33 pm showed sudden onset with Vs, V>A, rates 146-182 bpm, lasting 5 beats.
Therefore, there is no evidence of sustained VT or VF after ablation in the given medical note.
Therefore, the answer is No. |

Table 4: (continued)

| Case 3 (Yes VT Recurrence) |
| --- |
| Interrogation completed...The patient's underlying rhythm is SB...Summary Comments: Full ICD interrogation completed. ...The histogram demonstrated a normal heart rate profile...1 High ventricular event, showing NSVT, since last remote on ***. Patient had event with 2 shocks on ***. Patient had called and discussed with MD and he is here today for device adjustment. ... |

| SRR Response | Zero-Shot-CoT Response |
| --- | --- |
| rationale: 
 - 1 High ventricular event, showing NSVT, since last remote on ***. 
 - Patient had event with 2 shocks on ***. 
 label: **Yes** | Based on the given medical note, the patient has had an episode of nonsustained ventricular tachycardia (NSVT) since the last remote check on ***. The patient experienced this event with 2 shocks on ***. The patient called and discussed the event with the doctor, and is now here for device adjustment. The patient had an EKG done on the same day and it will be reviewed by Dr. ***. The note also mentions that there were no mode switch episodes and that the histogram demonstrated a normal heart rate profile. The summary comments state that the ICD interrogation was completed and that sensing, thresholds, and lead impedances are stable on both leads. The normal ICD function was confirmed and all findings were reviewed with Dr. ***. However, the note does not mention sustained ventricular tachycardia (VT) or ventricular fibrillation (VF) after ablation. It only mentions the episode of NSVT and the subsequent device adjustment. Therefore, based on the given information, it cannot be concluded whether the patient has had sustained VT or VF after ablation. Therefore, the answer is No. |

| Case 4 (No VT Recurrence) |
| --- |
| Cardiac Monitor Report. Reason for Study: Presence of Automatic. Type of cardiac monitor: Zio Patch. Interpretation Summary...Sinus Rhythm was observed. Average HR 77, Minimum 48, Maximum 176....PVC burden was: 3.8% There were 2 episodes of wide complex tachycardia lasting up to 6 beats with rates up to 141 bpm. Wide complex tachycardia appeared to be: Ventricular Tachycardia. There were 5 patient triggered events which show VT, Sinus, VEs... |

| SRR Response | Zero-Shot-CoT Response |
| --- | --- |
| rationale: 
 - There were 2 episodes of wide complex tachycardia lasting up to 6 beats with rates up to 141 bpm. 
 - Wide complex tachycardia appeared to be: Ventricular Tachycardia. 
 - There were 5 patient triggered events which show VT, Sinus, VEs. 
 label: Yes | Based on the given medical note, the patient has had episodes of wide complex tachycardia, which appeared to be ventricular tachycardia (VT). There were 2 episodes of wide complex tachycardia lasting up to 6 beats with rates up to 141 bpm. Additionally, there were 5 patient triggered events which show VT, Sinus, VEs. However, there is no mention of sustained ventricular tachycardia or ventricular fibrillation after ablation. The note primarily indicates predominantly sinus rhythm with no significant abnormalities detected. Therefore, the answer is **No**. |

Table 4: (continued)

| Case 5 (No VT Recurrence) |
|---|
| Cardiac Monitor Report. Reason for Study: Ventricular Premature. Type of cardiac monitor: Zio Patch...Interpretation Summary...Sinus Rhythm was observed. Average HR 65 bpm, Minimum 36 bpm, Maximum 190 bpm...There were 6 episodes of wide complex tachycardia lasting up to 10 beats with rates up to 190 bpm. Wide complex tachycardia appeared to be: Ventricular Tachycardia...There were 12 patient symptom episodes which correlate with Sinus, Ventricular Bigeminy, SVE(s), VE(s). IMPRESSION: Results demonstrate predominantly sinus rhythm. |

| SRR Response | Zero-Shot-CoT Response |
|---|---|
| rationale:
- There were 6 episodes of wide complex tachycardia lasting up to 10 beats with rates up to 190 bpm.
- Wide complex tachycardia appeared to be: Ventricular Tachycardia.
label: Yes | No, the patient has not had sustained ventricular tachycardia (VT) or ventricular fibrillation after ablation. Based on the given medical note, the patient has not had sustained ventricular tachycardia (VT) or ventricular fibrillation after ablation. Here's the breakdown:
1. Sinus Rhythm: The patient's cardiac monitor report shows sinus rhythm with an average heart rate of 65 bpm, minimum 36 bpm, and maximum 190 bpm. This indicates that the patient's heart rhythm is normal.
2. AV Block: The report states that AV Block (2nd° Mobitz II, 3rd°) was not present. This means that there were no abnormalities in the electrical conduction between the atria and ventricles.
3. Supraventricular Ectopic (SVE) beats: The report mentions that the SVE beat burden was less than 1.0%. This indicates that there were very few abnormal beats originating from above the ventricles.
4. SVT (Supraventricular Tachycardia): There were 57 episodes of SVT observed, with the fastest rate being 179 bpm. However, the SVT appeared to be atrial tachycardia, not ventricular tachycardia.
5. Atrial fibrillation and/or flutter: The report states that atrial fibrillation and/or flutter was not present. This means that there were no abnormal rapid and irregular heart rhythms originating from the atria.
6. PVC (Premature Ventricular Contractions) burden: The PVC burden was 9.5%, indicating that there were some premature beats originating from the ventricles.
7. Wide complex tachycardia: There were 6 episodes of wide complex tachycardia lasting up to 10 beats with rates up to 190 bpm. However, it appeared to be ventricular tachycardia, not sustained ventricular tachycardia.
8. Patient triggered events and symptoms: There were 12 patient triggered events and symptom episodes which correlated with sinus rhythm, ventricular bigeminy, SVE (s), and VE (s). This indicates that the patient experienced various heart rhythm abnormalities, but sustained ventricular tachycardia or ventricular fibrillation was not mentioned.
In conclusion, based on the given medical note, the patient has not had sustained ventricular tachycardia (VT) or ventricular fibrillation after ablation.
Therefore, the answer is No. |

Table 4: (continued)

| Case 6 (No VT Recurrence) |
| --- |
| Interrogation completed...The following therapy was delivered: 0 shocks and 0 ATP. Patient has had 0 episodes of VT and 0 episodes of VF detected....Summary Comments: Full BIV ICD interrogation completed...Review of old episode list showed 7 VT events recorded. Stored events showed VS>AS, rate at 160-176 bpm, and each episode terminated by ATP x 1. 5 NSVT events recorded. Stored EGMs showed VS>AS, rate at 163-179 bpm, and lasting 8-17 bpm. VT zone is at 110 bpm w/ NID of 18 beats. Pt had syncopal event on ***. No events were recorded on that day to correlate w/ syncope...There were NSVT and VT terminated by ATP events noted. |

| SRR Response | Zero-Shot-CoT Response |
| --- | --- |
| rationale:
- The following therapy was delivered: 0 shocks and 0 ATP.
- Patient has had 0 episodes of VT and 0 episodes of VF detected.
- There were NSVT and VT terminated by ATP events noted.
label: Yes | No, the patient has not had sustained ventricular tachycardia (VT) or ventricular fibrillation (VF) after ablation. Based on the given medical note, the patient has not had sustained ventricular tachycardia (VT) or ventricular fibrillation (VF) after ablation. The note states that the patient has had 0 episodes of VT and 0 episodes of VF detected. Additionally, the therapy delivered shows 0 shocks and 0 ATP (anti-tachycardia pacing) events.
However, it is important to note that the patient has a history of VT events. The old episode list showed 7 VT events recorded, with each episode terminated by ATP (anti-tachycardia pacing). There were also 5 episodes of nonsustained ventricular tachycardia (NSVT) recorded, with each episode lasting 8-17 beats and terminated by ATP.
Overall, the current device check shows normal device function, stable sensing, thresholds, and impedances. The patient has not experienced any sustained VT or VF after ablation, but there is a history of VT and NSVT events that have been successfully terminated by ATP. These findings should be reviewed by the attending physician, Dr. ***.
Therefore, the answer is No. |

# B  EVALUATION OF STRUCTURED RATIONALE RESPONSES (SRR) PROMPTING ON VARIOUS STRUCTURED FORMATS, LLMS, AND DATASETS

## B.1  OUR SRR CAN BE GENERALIZED TO DIFFERENT STRUCTURED FORMATS

We have expanded our experiments to evaluate the generalizability of our findings across various structured formats, including YAML (YAML Ain't Markup Language) and INI. The experimental outcomes are presented in Figure 7, while the detailed prompts are provided in Table 5.

Our SRR prompting consistently enhanced performance compared to the standard prompt, rationale, and standalone structured formats. Echoing the findings from Section 4.5, requesting structured output format significantly impacted performance. Notably, our SRR achieved an average balanced accuracy improvement of 4% with JSON (NRI=0.164, $p = 0.539$), 16% with YAML (NRI = 0.736, $p = 0.001$), and 7.3% with INI (NRI=0.322, $p = 0.181$), across three clinical questions.

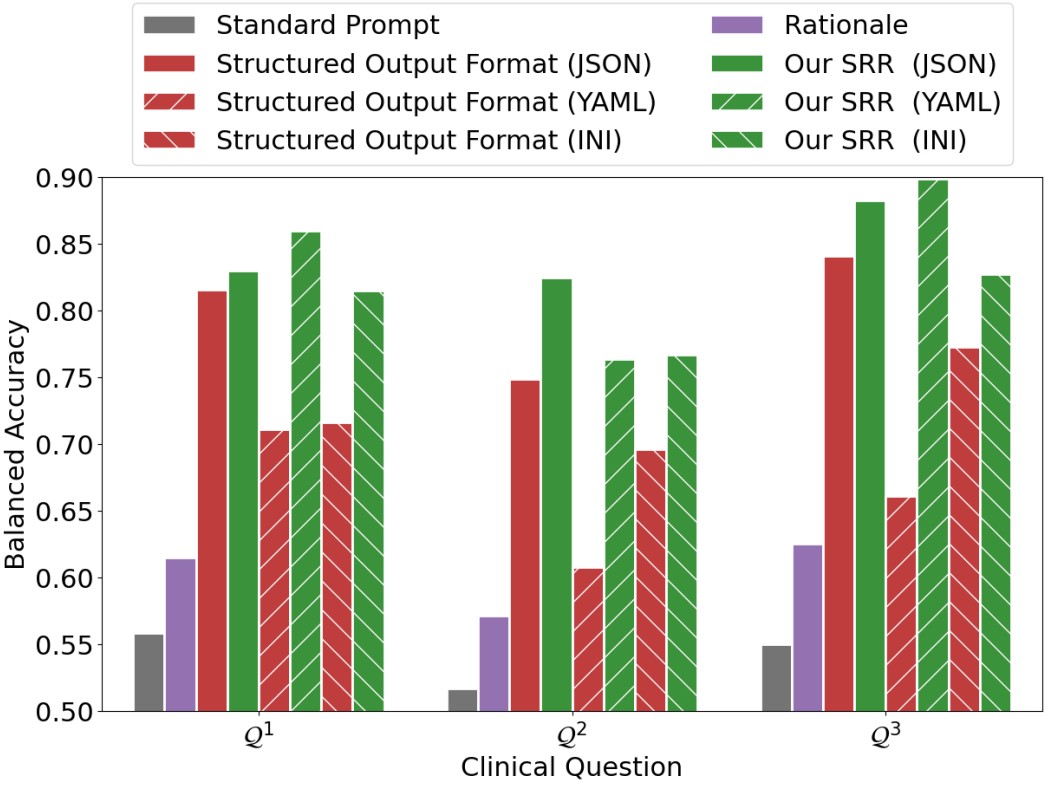

Figure 7: Performance comparison of SRR with different structured formats.

Table 5: Prompt Definitions of our SRR with YAML and INI

| **Our SRR Prompt with YAML** |
|---|
| ### Question:
Based on the given medical note as input, <Clinical Question $\mathcal{Q}^1$, $\mathcal{Q}^2$, or $\mathcal{Q}^3$ Here>.
### Input:
<Test Note Here>
### Instruction:
Above is a question, paired with an input that provides further context. Write a response in YAML format that answers the question. The response should contain two key-value pairs with two keys: "rationale" and "label". The value of the "rationale" key should be a bulleted list of sentences supporting the answer of the "label". Each sentence should be a quote from the input medical note. The value of the "label" key should be "Yes" or "No". Your response should only contain the YAML string.
### Response: |
| **Structured Output Format Prompt with YAML** |
| ### Question:
<Same as Standard Prompt>
### Input:
<Test Note Here>
### Instruction:
Above is a question, paired with an input that provides further context. Write a response in YAML format that answers the question. The response should contain a single key-value pair. The key should be "label", and the value should be either "Yes" or "No". Your response should only contain the YAML string.
### Response: |
| **Our SRR Prompt with INI** |
| ### Question:
Based on the given medical note as input, <Clinical Question $\mathcal{Q}^1$, $\mathcal{Q}^2$, or $\mathcal{Q}^3$ Here>.
### Input:
<Test Note Here>
### Instruction:
Above is a question, paired with an input that provides further context. Write a response in INI format that answers the question. The response should contain a section named "Response". Inside this section, there should be two properties: "rationale" and "label". The value of the "rationale" property should contain a bulleted list of sentences supporting the answer of the "label". Each sentence should be a quote from the input medical note, separated by semicolons. The value of the "label" property should be "Yes" or "No". Your response should only contain the INI string.
### Response: |
| **Structured Output Format Prompt with INI** |
| ### Question:
<Same as Standard Prompt>
### Input:
<Test Note Here>
### Instruction:
Above is a question, paired with an input that provides further context. Write a response in INI format that answers the question. The response should contain a section named "Response". Inside this section, there should be a property "label" with a value of either "Yes" or "No". Your response should only contain the INI string.
### Response: |

## B.2 Our SRR uniformly enhanced the response accuracy across different LLMs

To broaden our evaluation, we sought an LLM fundamentally different from GPT-3.5-turbo, thereby excluding GPT-4. Furthermore, to ensure that our data would not be utilized for training or tuning purposes, we excluded models like PaLM, as per the terms outlined by Google[2].

Consequently, we leveraged the Amazon Bedrock platform to conduct experiments using Jurassic-2 from AI21 Labs, one of the largest available LLMs on the platform. We repeated the experiments with the standard prompt, Zero-Shot-CoT (Kojima et al., 2022), 5-shot prompting, and our SRR. The results are summarized in the Figure 8

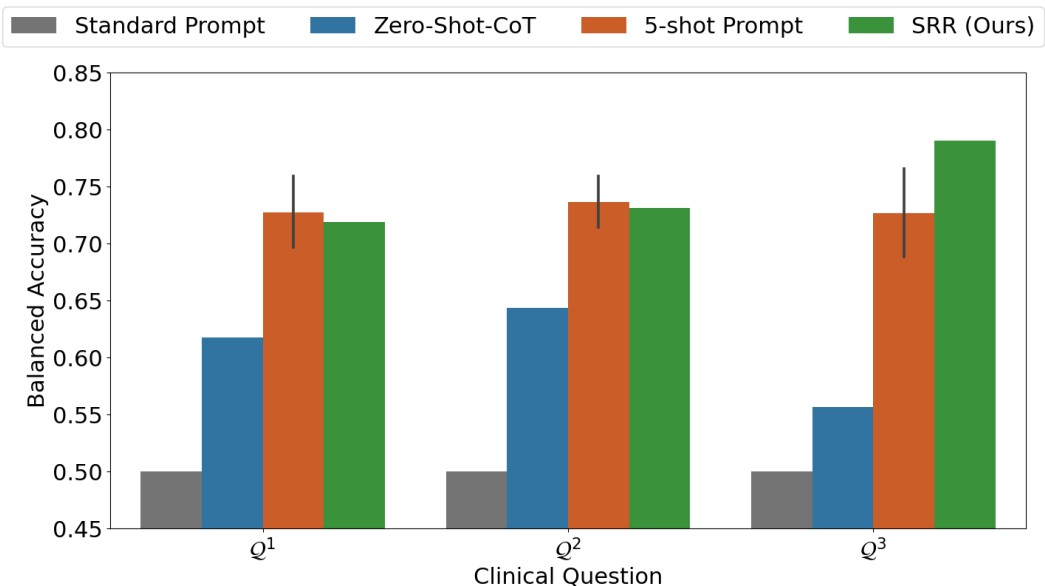

Figure 8: Performance comparison of different prompting strategies using Jurassic-2 accessed through Amazon Bedrock. Our SRR strategy outperformed the baseline prompting strategies across three clinical questions, demonstrating an average balanced accuracy improvement of 24.7% over the standard prompt (NRI=1.131, $p < 0.001$), 14.0% over the Zero-Shot-CoT prompt (NRI=0.925, $p < 0.001$), and comparable performance to the 5-shot prompt (NRI=-0.125, $p = 0.640$).

Compared to GPT-3.5-turbo, Jurassic-2 exhibited reduced capabilities with the standard prompt, consistently returning a "No" response irrespective of the input notes, thereby failing to detect any VT recurrence cases. This lower baseline performance significantly impeded Zero-Shot-CoT's effectiveness, emphasizing its limitations when the LLM is in the absence of requisite domain knowledge. In contrast, our SRR method still markedly enhanced performance, significantly outperforming Zero-Shot-CoT ($p < 0.001$) and achieving competitive results compared to the 5-shot prompt, but without necessitating expert inputs.

### B.3  OUR SRR ACHIEVED ENHANCED EFFICIENCY AND COMPETITIVE PERFORMANCE IN A LESS NUANCED CLINICAL TASK

We extended our evaluation to include the *public* EHR dataset from the 2006 n2c2 De-identification and Smoking Status Challenge, focusing on identifying patient smoking status from medical discharge records (Uzuner et al., 2008). The dataset comprised 398 training and 104 test records. These records were classified into one of the five pre-defined categories: past smoker, current smoker, smoker, non-smoker, and unknown.

Table 6: Prompt Definitions on the smoking status identification task from the 2006 n2c2 Deidentification and Smoking Status Challenge (Uzuner et al., 2008).

**Standard Prompt**
### Question:
Based on the given medical discharge summary as input, categorize the smoking status of the patient into one of the following five categories: Past Smoker, Current Smoker, Smoker, Non-Smoker, Unknown.
[Definitions]
Past Smoker: A patient who has not smoked for at least one year but was a smoker more than a year ago.
Current Smoker: A patient who was a smoker within the past year.
Smoker: A patient who has a history of smoking (either current or past) but lacks sufficient information to be classified as a Past Smoker or Current Smoker.
Non-Smoker: A patient with a discharge summary indicating they have never smoked.
Unknown: A patient whose discharge summary does not provide information about their smoking status, or where it's unclear whether they are a Current or Past Smoker.
### Input:
<Test Note Here>
### Response:

**Zero-shot-CoT Prompt**
### Question:
<Same as Standard Prompt>
### Input:
<Test Note Here>
### Response:
Let's think step by step.

**Our SRR Prompt**
### Question:
<Same as Standard Prompt>
### Input:
<Test Note Here>
### Instruction:
Above is a question, paired with an input that provides further context. Analyze the input text based on the definitions and write a response in JSON format that answers the question. The response should be a dictionary with two keys: "rationale" and "label". The value of the "rationale" key should be a bulleted list of sentences supporting the answer of the "label". Each sentence should be a quote from the input medical note. The value of the "label" key should be "Past Smoker", "Current Smoker", "Smoker", "Non-Smoker", or "Unknown". Your response should only contain the JSON string.
### Response:

We conducted experiments using GPT-3.5-turbo and designed a standard prompt based on the task description, and further augmented it with Zero-shot-CoT (Kojima et al., 2022) and our Structured Rationale Responses (SRR). We detailed each prompt in Table 6.

The length of the records ($1144.68 \pm 718.77$ tokens) precluded the use of few-shot prompting. We also excluded one note from both the training and test data (ID=909 and ID=709, respectively) since the Zero-shot-CoT and our SRR prompt with these two notes exceed the input length limitation. The performance metrics, as per Uzuner et al. (2008), include macroaveraged and microaveraged F1 scores, as well as Cohen's kappa ($\kappa$) between LLM responses and ground truth. The results are presented in Table 7.

Without specialized prompting, the LLM struggled to correctly identify smoking status from complex medical discharge records. Unlike VT recurrence, which is nuanced by diverse clinical synonyms, smoking status detection, being more straightforward and reliant on general knowledge, aligns well with the capabilities of Zero-Shot-CoT (Kojima et al., 2022), which markedly outper-

---
[2]PaLM API and MakerSuite Additional Terms of Service

formed the standard prompt. Notably, our SRR approach yielded competitive performance to Zero-Shot-CoT, with a higher macroaveraged F1 score and $\kappa$, indicating a more balanced performance across minority classes, a point elaborated in Section 2.3 of our paper. Importantly, our method also offered a significant reduction in cost and time (approximately 50%), by requiring only a single inquiry as opposed to the two-step process required by Zero-Shot-CoT.

Table 7: Performance comparison of different prompting strategies on the smoking status identification task from the 2006 n2c2 De-identification and Smoking Status Challenge (Uzuner et al., 2008). Cohen's kappa ($\kappa$) is calculated between LLM responses and ground truth.

| | Training Data | | | Test Data | | |
|---|---|---|---|---|---|---|
| | F1 (Macro) | F1 (Micro) | $\kappa$ | F1 (Macro) | F1 (Micro) | $\kappa$ |
| Standard Prompt | 0.16 | 0.24 | 0.26 | 0.33 | 0.48 | 0.29 |
| Zero-Shot-CoT | 0.58 | **0.78** | 0.57 | 0.50 | **0.75** | 0.51 |
| SRR (Ours) | **0.63** | 0.73 | **0.60** | **0.57** | 0.66 | **0.52** |

