# OpenReview forum: "Optimized Large Language Models Accurately Identify Recurrence of VT After Ablation from Complex Medical Notes: Will Chart Review Become Obsolete?"
_ICLR.cc/2024/Conference — ICLR 2024 Conference Withdrawn Submission_

### Official Review · Reviewer_NLwC · 2023-10-31

**Soundness:** 2 fair
**Presentation:** 2 fair
**Contribution:** 2 fair
**Rating:** 3
**Confidence:** 4

**Summary:**

This paper reports on a clinical language processing application of large language models in English.

**Strengths:**

Considering large language models is a trending topic. In particular, prompt engineering is quite well done in the paper.

Clinical language processing is topic of utmost societal importance, and the particular application to Ventricular Tachycardia (VT) has not been really addressed, even for English.

This study is sensitive to its clinical context, as evidenced by its there research questions, and the large scale of clinical data used in the experiments of this study is excellent.

The manuscript is very carefully written.

**Weaknesses:**

The study could be strengthened by addressing, e.g., ethical code of conduct with respect to involving clinical experts as annotators in this study and the risks of using ChatGPT to process clinical notes. This could form a broader impact statement in the discussion section. More generally, the study is lacking in its discussion (e.g., limitations of this study and proposed future work).

Although performance evaluation results seem great, I cannot find statistical significance testing, confidence intervals, or similar included in the study methodology.

The paper could be made stronger by being more specific about the language(s) considered. It is implied that it is limited to English only at its current form.

**Questions:**

What is the broader impact of this work? What kind of benefits and risks did it have? Were the findings statistically and/or practically significant? Have related work been conducted for languages other than English? What the envisioned influence and impact will be? What kind of future work should be conducted?

**Details Of Ethics Concerns:**

No concerns. Research ethics were quite well addressed, although, as commented above, potentially harmful insights, methodologies, and applications could be elaborated as a broader impact statement.

---

> ### Author Response · Authors · 2023-11-22
> **Enhanced Manuscript with Extended Discussions on Ethics, Risks, Limitations and Future Work**
>
> We express our sincere gratitude for the reviewer's positive remarks regarding the quality of our prompt engineering, and the acknowledgment of the significance of clinical language processing, particularly in its application to Ventricular Tachycardia (VT). We are also thankful for the reviewer's appreciation of the careful writing of our manuscript and the encouraging feedback on the experimental results.
>
> ## Comment 1
> The study could be strengthened by addressing, e.g., ethical code of conduct with respect to involving clinical experts as annotators in this study and the risks of using ChatGPT to process clinical notes. This could form a broader impact statement in the discussion section. More generally, the study is lacking in its discussion (e.g., limitations of this study and proposed future work).
>
> ## Response 1
> We are thankful for the reviewer's insightful feedback highlighting the need for a more comprehensive discussion in our study. In response, we have extensively revised Section 6 of our manuscript to include these crucial aspects:
>
> **Ethical Considerations**: We have included a detailed discussion on the ethical code of conduct employed in our study, particularly concerning the involvement of clinical experts in the manual annotation of clinical data. This discussion emphasizes the measures taken to ensure ethical compliance and the integrity of the data annotation process.
>
> **Risks of Using LLMs**: The risks associated with using LLMs to process clinical data have been extensively explored. We specifically address concerns regarding privacy leakage and outline potential practices to mitigate such risks. Additionally, we delve into the challenges posed by the "black-box" nature of LLMs and their potential for "hallucination." Here, we also highlight the advantages of our SRR prompting in providing the precise rationale, contributing to more reliable and interpretable outputs.
>
> **Limitations and Future Work**: The section further elaborates on the limitations of our current study and maps out potential avenues for future work. This includes expanding our dataset, evaluating domain-specific models pre-trained with EHR data, and exploring the application of our methodology in multilingual corpora.
>
> We believe that these comprehensive additions and discussions significantly enhance the depth and breadth of our work, effectively addressing the concerns raised by the reviewer.
>
> For detailed discussions, please refer to Section 6 in the revised version of our manuscript.
>
> ## Comment 2
> The paper could be made stronger by being more specific about the language(s) considered. It is implied that it is limited to English only at its current form.
>
> ## Response 2
> We are grateful for the reviewer's insightful suggestion regarding the evaluation of our methodology in multiple languages.
>
> In the current scope of our study, our evaluation was primarily conducted using English language texts, constrained largely by the availability of relevant datasets. While we acknowledge that the specifics of our results may differ across different languages, the fundamental intuition behind our SRR strategy will likely hold. Adapting our evaluation framework to include corpora in diverse languages is not only a feasible aim but one that can significantly broaden the scope of our research. This expansion is enabled by the inherent multilingual capabilities of modern LLMs, representing a significant advantage over traditional NLP methodologies.
>
> These considerations and potential extensions have been discussed in detail in Section 6 of our revised manuscript.

---

> > ### Author Response · Authors · 2023-11-22
> > **Inclusion of Statistical Significance Testing**
> >
> > ## Comment 3
> > Although performance evaluation results seem great, I cannot find statistical significance testing, confidence intervals, or similar included in the study methodology.
> >
> > ## Response 3
> > We are thankful to the reviewer for highlighting the absence of statistical significance testing in our study.
> >
> > In the revised manuscript, we employed the Net Reclassification Index (NRI), as described by Pencina et al. (2008), to conduct statistical comparisons between different prompting strategies. This method is particularly apt for classification tasks involving heavily unbalanced classes, such as in our dataset, which comprises only 7% VT cases.
> >
> > Additionally, we opted for a fixed split approach (detailed in Table 1) over cross-validation. This methodological choice, necessitated by the limited number of VT cases in our dataset, is elaborated in our Response 3 to Reviewer T6RQ. Moreover, considering that the LLMs utilized were not trained or tuned with our data, the responses for each defined prompt were deterministic. This precluded the inclusion of confidence intervals in our analysis.
> >
> > We hope these clarifications address the reviewer's concerns and provide a comprehensive understanding of our methodological choices.
> >
> > Reference
> >
> > Michael J Pencina, Ralph B D’Agostino Sr, Ralph B D’Agostino Jr, and Ramachandran S Vasan. Evaluating the added predictive ability of a new marker: from area under the roc curve to reclassification and beyond. Statistics in medicine, 27(2):157–172, 2008.
> >
> > ## Summary
> >
> > We are deeply grateful for the constructive feedback from the reviewers, which has been instrumental in substantially enhancing the depth, breadth, and rigor of our study.

---

### Official Review · Reviewer_T6RQ · 2023-11-01

**Soundness:** 2 fair
**Presentation:** 3 good
**Contribution:** 3 good
**Rating:** 6
**Confidence:** 3

**Summary:**

The paper proposed a non-domain-specific prompting strategy, Structured Rationale Responses (SRR) to enhance the accuracy of LLM responses to identify recurrence of Ventricular Tachycardia (VT) among electronic health records (EHRs).

**Strengths:**

- Originality: the paper proposed a creative combination of existing prompting strategies with novel application to VT identification from EHR data.
- Clarity: the paper is well-organized with clear description in study design and method illustration, as well as results examples.

**Weaknesses:**

1. The proposed method is only evaluated by a single dataset for a single disease VT, which might weaken the utility of the proposed prompting strategy that is VT-specific only, but not generalized to other diseases.
2. The experiment results lack the model comparison of using other structured formats in the prompt, e.g. xml or others. What exact prompt was used with bullet points in experiment 4.5?
3. Figure 3, the experiment results of SRR lack confidence bars. It is the same in Figure 5 and 6. Is the results cross-validated? If not, how robust is the performance.
4. Lack of limitation discussion in the conclusion part.

**Questions:**

1. How will the proposed prompting strategy work in GPT-4?

---

> ### Author Response · Authors · 2023-11-21
> **Our SRR can be generalized to different structured formats**
>
> We are grateful for the reviewers' comments acknowledging that our method is "creative", and our paper is "well-organized with clear description."
>
> ## Comment 1:
> The experiment results lack the model comparison of using other structured formats in the prompt, e.g. xml or others.
>
> ## Response 1:
> We thank the reviewer for this insightful suggestion.
>
> In response, we have expanded our experiments to evaluate the generalizability of our findings across various structured formats, including YAML (YAML Ain't Markup Language) and INI. The results of these experiments are summarized below:
>
> | **Method**          | $\mathcal{Q}^1$ | $\mathcal{Q}^2$ | $\mathcal{Q}^3$ |
> |---------------------|-----------------|-----------------|-----------------|
> | **Standard Prompt** | 0.56            | 0.52            | 0.55            |
> | **Rationale**       | 0.61            | 0.57            | 0.63            |
> | **JSON**            | 0.82            | 0.75            | 0.84            |
> | **SRR with JSON**   | 0.83     | 0.82   | 0.88   |
> | **YAML**            | 0.71            | 0.61            | 0.66            |
> | **SRR with YAML**   | 0.80   | 0.76  | 0.90   |
> | **INI**             | 0.72            | 0.70            | 0.77            |
> | **SRR with INI**    | 0.81   | 0.77   | 0.83   |
>
> Our SRR prompting consistently enhanced performance compared to the standard prompt, rationale, and standalone structured formats. Echoing the findings from Section 4.5, requesting structured output format significantly impacted performance. Notably, our SRR achieved an average balanced accuracy improvement of 4% with JSON (NRI=0.164, $p=0.539$), 16% with YAML (NRI = 0.736, $p=0.001$), and 7.3% with INI (NRI=0.322, $p=0.181$), across three clinical questions.
>
> *NRI: Net Reclassification Index (NRI) (Pencina et al., 2008)
>
> Additionally, we experimented with the XML (eXtensible Markup Language) format, as the reviewer suggested. However, the performance of the LLM deteriorated significantly in this case, consistently returning a "No" response for all queries. This failure might be attributed to XML’s inherent complexity, potentially leading the LLM to prioritize format accuracy over the correctness of the response.
>
> These results, along with a detailed discussion, have been added to Section B.1 of our revised manuscript.

---

> > ### Author Response · Authors · 2023-11-21
> > **Response to Other Comments**
> >
> > ## Comment 2:
> > The proposed method is only evaluated by a single dataset for a single disease VT, which might weaken the utility of the proposed prompting strategy that is VT-specific only, but not generalized to other diseases.
> >
> > ## Response 2:
> > We appreciate the reviewer’s observation regarding the scope of our evaluation being limited to a single dataset focusing on Ventricular Tachycardia (VT). Our primary research is concentrated on life-threatening heart rhythm disorders, such as ventricular arrhythmias, for which publicly available EHR datasets are scarce.
> >
> > Nevertheless, to address the reviewer's concerns and demonstrate the broader applicability of our prompting strategy, we have extended our evaluation to include an additional task: identifying smoking status from a publicly available EHR dataset. This expansion into a different clinical task illustrates the generalizability of our approach.
> >
> > For detailed results and a thorough discussion of this evaluation, please refer to our Response 4 to Reviewer QmtS, as well as Section B.3 in the revised version of our manuscript.
> >
> > ## Comment 3:
> > Figure 3, the experiment results of SRR lack confidence bars. It is the same in Figure 5 and 6. Is the results cross-validated? If not, how robust is the performance.
> >
> > ## Response 3:
> > We thank the reviewer for this comment. In our study, we opted for a fixed data split (as detailed in Table 1) over cross-validation. The primary challenge of cross-validation in our dataset was the small number of VT recurrence cases, which made it difficult to create several folds while maintaining a balanced distribution of VT and non-VT classes, along with other baseline characteristics outlined in Table 1.
> >
> > Despite this methodological choice, we have conducted extensive evaluations to affirm the efficacy of our SRR, spanning different clinical questions, structured formats, LLMs, and datasets. We hope this comprehensive testing has addressed any concerns regarding the robustness of our approach.
> >
> > ## Comment 4:
> > Lack of limitation discussion in the conclusion part.
> >
> > ## Response 4:
> > We thank the reviewer for pointing out the absence of a limitations discussion in our manuscript. In response, we have thoroughly revised the manuscript to include a comprehensive section (Section 6) discussing the limitations of our study and outlining the potential avenues for future work, including the expansion of our dataset and further evaluations using EHR-pretrained language models. We believe that this addition significantly enhances the depth and completeness of our manuscript.
> >
> > ## Comment 5:
> > How will the proposed prompting strategy work in GPT-4?
> >
> > ## Response 5:
> > We appreciate the reviewer's interest in the applicability of SRR with different LLMs. Considering that GPT-4 is a successor to GPT-3.5-turbo from the same provider (OpenAI) and may share similar underlying properties, we chose to use a fundamentally different LLM from an alternate provider, enabling a more diverse and robust evaluation.
> >
> > Detailed results and a thorough discussion of this evaluation can be found in our Response 2 to Reviewer QmtS, as well as Section B.2 in the revised version of our manuscript.
> >
> > ## Comment 6:
> > What exact prompt was used with bullet points in experiment 4.5?
> >
> > ## Response 6:
> > We appreciate the reviewer's attention to detail regarding the prompt used in experiment 4.5.
> >
> > We have added Section A.2 to define and elucidate the prompts used in our experiments clearly. We hope this expansion can clarify any confusion and provide a clearer understanding of the methodologies employed in our study.
> >
> > ## Summary
> > We hope that these revisions and experimental results satisfactorily address the concerns raised.

---

### Official Review · Reviewer_QmtS · 2023-11-02

**Soundness:** 2 fair
**Presentation:** 2 fair
**Contribution:** 1 poor
**Rating:** 3
**Confidence:** 3

**Summary:**

The paper introduces a novel prompting strategy called Structured Rationale Responses (SRR), which is specifically tailored to improve the precision and dependability of Language Model (LLM) responses when dealing with nuanced queries in non-publicly accessible electronic health records (EHR). SRR serves as a guidance mechanism for LLMs, encouraging them to produce responses that are 1) structured, such as in JSON format, and 2) accompanied by clear rationales.

**Strengths:**

1. The suggested approach is straightforward.
2. Ablation studies have been conducted, and the method has been compared to other widely used strategies.

**Weaknesses:**

My dissatisfaction with the evaluation of the proposed approach stems from several factors.

1. The authors argue for the superiority of the proposed strategy, but the reasoning behind this claim is not adequately explained. The analysis is limited to a single dataset addressing a specific problem, which is insufficient to draw conclusive results. It remains uncertain whether this strategy would be effective for non-private EHR data. Why exactly it is good for private EHR data?
2. The evaluation involves only one model, whereas related work typically considers a subset of models for a more comprehensive assessment.
3. The experimental results do not sufficiently clarify why this strategy is advantageous. For example, it's unclear whether model size plays a significant role in its effectiveness. Furthermore, an error analysis comparing the proposed strategy to other strategies could provide valuable insights.
4. Minor Issue: The use of the superscript $t$ is not defined or explained in the context of the text.

**Questions:**

Please, elaborate on Weakness section

---

> ### Author Response · Authors · 2023-11-21
> **Response to Reviewer Comments**
>
> We are grateful for the reviewer's recognition that our method is "straightforward" and acknowledgment of our efforts to compare it with "widely used strategies."
>
> ## Comment 1:
> The authors argue for the superiority of the proposed strategy, but the reasoning behind this claim is not adequately explained.
>
> The experimental results do not sufficiently clarify why this strategy is advantageous. For example, it's unclear whether model size plays a significant role in its effectiveness. Furthermore, an error analysis comparing the proposed strategy to other strategies could provide valuable insights.
>
> ## Response 1:
> We are grateful for the insightful feedback. In response, we have expanded our manuscript with **a new discussion in Section 2.3 that delves into the intuition behind our proposed SRR**, particularly its superiority over other reasoning-based prompting methods.
>
> We added **extensive experimental evaluation** with different structured formats, LLMs, and tasks (Sections B.1, B.2, and B.3), to further clarify the effectiveness and robustness of our proposed SRR.
>
> We also added Section A.4, presenting a **comprehensive error analysis** comparing SRR with other prompting strategies. This analysis offers valuable insights into the effectiveness and limitations of our method.
>
> For your convenience, the key points from the newly added paragraph in Section 2.3 are as follows:
>
> *LLMs often lack adequate domain-specific clinical knowledge, resulting in unreliable outcomes when LLMs are compelled to reason independently. The issue of forced reasoning is further highlighted by Zero-shot-CoT’s diminished performance with less sophisticated LLMs (Section B.2).*
>
> *In contrast, our Rationale-based intervention, which emphasizes contextual cue identification over explicit deduction, successfully provided the correct answer (Figure 1B). Our approach might lean towards a ``Yes" classification in cases of uncertainty, potentially increasing false positives (refer to Section A.4). Despite this, our method's superior ability to identify the minority but critical classes demonstrates its reliability. This attribute is particularly valuable in health applications with highly skewed category distributions. Additionally, requesting structured formatting can diminish ambiguity, allowing LLMs to discern the rationale behind inquiries more rapidly, leading to more effective and efficient computational resource allocation, which is crucial for nuanced tasks.*
>
> ## Comment 2:
> The use of the superscript $t$ is not defined or explained in the context of the text.
>
> ## Response 2:
> We added the following sentence in Section 2.1 to explain the superscript $t$:
>
> *The superscript $t$ indicates that the data belongs to the Test cohort $\mathcal{T}$.*

---

> > ### Author Response · Authors · 2023-11-21
> > **Our SRR uniformly enhanced the response accuracy across different LLMs**
> >
> > ## Comment 3:
> > The evaluation involves only one model, whereas related work typically considers a subset of models for a more comprehensive assessment.
> >
> > ## Response 3:
> > We appreciate the reviewer's emphasis on the necessity of evaluating our approach across different LLMs.
> >
> > To broaden our evaluation, we sought an LLM fundamentally different from GPT-3.5-turbo, thereby excluding GPT-4 due to its similar lineage. Furthermore, to ensure that our data would not be utilized for training or tuning purposes, we excluded models like PaLM, as per the terms outlined by Google (https://developers.generativeai.google/terms).
> >
> > Consequently, we leveraged the Amazon Bedrock platform to conduct experiments using Jurassic-2 from AI21 Labs, one of the largest available LLMs on the platform. We repeated the experiments with the standard prompt, Zero-Shot-CoT, 5-shot prompting, and our SRR. The results are summarized in the table below:
> >
> > |                | $\mathcal{Q}^1$ | $\mathcal{Q}^2$ | $\mathcal{Q}^3$ |
> > |---------------------|-----------------|-----------------|-----------------|
> > | **Standard Prompt** | 0.50            | 0.50            | 0.50            |
> > | **Zero-Shot-CoT**   | 0.62            | 0.64            | 0.56            |
> > | **5-shot Prompt**   | 0.75$\pm$0.06   | 0.75$\pm$0.05   | 0.74$\pm$0.08   |
> > | **SRR (Ours)**      | 0.72            | 0.73            | 0.79            |
> >
> > Compared to GPT-3.5-turbo, Jurassic-2 exhibited reduced capabilities with the standard prompt, consistently returning a "No" response irrespective of the input notes, thereby failing to detect any VT recurrence cases. This lower baseline performance significantly impeded Zero-Shot-CoT's effectiveness, emphasizing its limitations when the LLM is in the absence of requisite domain knowledge. In contrast, our SRR method still markedly enhanced performance, significantly outperforming Zero-Shot-CoT ($p<0.001$) and achieving competitive results compared to the 5-shot prompt, but without necessitating expert inputs.
> >
> > These results, along with a detailed discussion, have been added to Section B.2 of our revised manuscript.

---

> ### Author Response · Authors · 2023-11-21
> **Our SRR achieved enhanced efficiency and competitive performance in a less nuanced clinical task**
>
> ## Comment 4:
>
> The analysis is limited to a single dataset addressing a specific problem, which is insufficient to draw conclusive results.
>
> It remains uncertain whether this strategy would be effective for non-private EHR data. Why exactly it is good for private EHR data?
>
> ## Response 4:
> We appreciate the reviewer's critical observation regarding the limitation of our study being evaluated on a single dataset and the question about the applicability of our approach to non-private EHR data.
>
> Firstly, we acknowledge the scarcity of publicly available EHR datasets as noted by Wornow et al. (2023). Additionally, datasets like MIMIC-III/IV, which are publicly available, cannot be shared with LLM APIs provided by companies like OpenAI due to data privacy concerns (https://physionet.org/news/post/415). Moreover, as elaborated in a recent review (Gao et al., 2022), the available datasets do not align well with our research focusing on life-threatening heart rhythm disorders, such as ventricular arrhythmias. Consequently, we concentrated our evaluation on an in-house dataset.
>
> Nonetheless, in response to Reviewers QmtS and T6RQ, we extended our evaluation to include the public EHR dataset from the n2c2 De-identification and Smoking Status Challenge 2006, focusing on identifying patient smoking status from medical discharge records (Uzuner et al., 2008). We conducted experiments using GPT-3.5-turbo and designed a standard prompt based on the task description, and further augmented it with Zero-shot-CoT and our Structured Rationale Responses (SRR). The length of the records (1144.68 $\pm$ 718.77 tokens) precluded the use of few-shot prompting. The performance metrics, as per Uzuner et al. (2008), include macroaveraged and microaveraged F1 scores, as well as Cohen's kappa ($\kappa$) between LLM responses and ground truth. The results are summarized in the table below:
>
> |             |            | Training Data |      |            | Test Data  |       |
> |---------------------|------------|---------------|------|------------|------------|-------|
> |             | F1 (Macro) | F1 (Micro)    | $\kappa$    | F1 (Macro) | F1 (Micro) | $\kappa$     |
> | **Standard Prompt** | 0.16       | 0.24          | 0.26 | 0.33       | 0.48       | 0.29  |
> | **Zero-Shot-CoT**   | 0.58       | 0.78          | 0.57 | 0.50       | 0.75       | 0.51  |
> | **SRR (Ours)**      | 0.63       | 0.73          | 0.60 | 0.57       | 0.66       | 0.52  |
>
> Without specialized prompting, the LLM struggled to correctly identify smoking status from complex medical discharge records. Unlike VT recurrence, which is nuanced by diverse clinical synonyms, smoking status detection, being more straightforward and reliant on general knowledge, aligns well with the capabilities of Zero-Shot-CoT, which markedly outperformed the standard prompt. Notably, our SRR approach yielded competitive performance to Zero-Shot-CoT, with a higher macroaveraged F1 score and $\kappa$, indicating a more balanced performance across minority classes, a point elaborated in Section 2.3 of our paper. Importantly, our method also offered a significant reduction in cost and time (approximately 50\%), by requiring only a single inquiry as opposed to the two-step process required by Zero-Shot-CoT.
>
>
> These results, along with a detailed discussion, have been added to Section B.3 of our revised manuscript.
>
> Reference
>
> Michael Wornow, Yizhe Xu, Rahul Thapa, Birju Patel, Ethan Steinberg, Scott Fleming, Michael A  Pfeffer, Jason Fries, and Nigam H Shah. The shaky foundations of large language models and foundation models for electronic health records. npj Digital Medicine, 6(1):135, 2023.
>
> Yanjun Gao, Dmitriy Dligach, Leslie Christensen, Samuel Tesch, Ryan Laffin, Dongfang Xu, Timothy Miller, Ozlem Uzuner, Matthew M Churpek, and Majid Afshar. A scoping review of publicly available language tasks in clinical natural language processing. Journal of the American Medical Informatics Association, 29(10):1797–1806, 2022.
>
> Ozlem Uzuner, Ira Goldstein, Yuan Luo, and Isaac Kohane. Identifying patient smoking status from medical discharge records. Journal of the American Medical Informatics Association, 15 (1):14–24, 2008.

---

> > ### Author Response · Authors · 2023-11-21
> > **Summary**
> >
> > We hope you find our responses, accompanied by comprehensive experimental evaluations and detailed discussions, satisfactory.
> >
> > Thank you once again for the valuable feedback.

---

### Comment · Reviewer_NLwC · 2023-11-20
**Summary**

Based on the three reviews and no author response, I cannot recommend accepting the paper. I do not see a need to change my original review either.

---

> ### Author Response · Authors · 2023-11-20
> **Response to Reviewer Feedback: Ongoing Revisions and Additional Experiments**
>
> Dear Reviewer NLwC:
>
> Thank you for your valuable feedback on our manuscript. We greatly appreciate the time and effort you have invested in reviewing our work.
>
> We are currently in the process of extensively revising our manuscript to address the concerns and suggestions you and other reviewers have raised. As part of this revision, we are adding $\textit{more discussions}$ related to the ethical code of conduct, limitations, and possible future work and conducting $\textit{new experiments}$ to provide additional evidence. We believe these efforts will significantly strengthen the manuscript.
>
> We anticipate that the revised version of our manuscript, as well as detailed responses, will be ready for submission by $\textbf{11/20}$ (End of Day, Anywhere on Earth). $\textbf{And we look forward to your constructive feedback on our resubmitted manuscript}$.
>
> Sincerely,

---

### Author Response · Authors · 2023-11-21
**Summary of Response**

We would like to express our sincere appreciation for the time and effort the reviewers have taken to evaluate our manuscript.

The constructive comments and insights have been extremely valuable in enhancing the quality and clarity of our work.

We have thoroughly addressed their questions and made necessary amendments to our manuscript in accordance with their recommendations. Each change in the manuscript has been marked in blue for your convenience.

$\textbf{Summary of main changes}$

1. Added a new paragraph in Section 2.3 exploring the intuition behind our proposed SRR, in response to comments from Review QmtS.
2. Conducted an in-depth error analysis in Section A.4, following the suggestion of Review QmtS.
2. Incorporated statistical significance testing, as commented by Review NLwC.
3. Performed a new evaluation with different structured formats (Section B.1), as Review T6RQ commented.
4. Performed a new evaluation with a different LLM (Section B.2), as Reviews QmtS and T6RQ commented.
5. Performed a new evaluation with a different task on a public EHR dataset (Section B.3), as Reviews QmtS and T6RQ commented.
6. Added a comprehensive discussion including ethical concerns, risks, limitations, and future work in Section 6, as Reviews T6RQ and NLwC advised.

We firmly believe that these revisions have significantly improved the breadth and depth of our research study.

We appreciate the opportunity to revise our manuscript and hope that the changes meet with your approval.